# Targeting SOX13 inhibits assembly of respiratory chain supercomplexes to overcome ferroptosis resistance in gastric cancer

Hui Yang [1,2,3,16], Qingqing Li [2,4,16], Xingxing Chen [5,16], Mingzhe Weng [6,16], Yakai Huang[7,16], Qiwen Chen[8], Xiaocen Liu [2,9,10], Haoyu Huang[1,2,10,11], Yanhuizhi Feng[12], Hanyu Zhou [1,2,10,11], Mengying Zhang[1,2,10,11], Weiya Pei[1,2,10,11], Xueqin Li[1,2,10,11], Qingsheng Fu [13], Liangyu Zhu [1,2,10], Yingying Wang[9], Xiang Kong [1,2,10,11], Kun Lv [1,2,10,11] ✉, Yan Zhang [2,14] ✉, Yangbai Sun[15] ✉ & Mingzhe Ma [7] ✉

Therapeutic resistance represents a bottleneck to treatment in advanced gastric cancer (GC). Ferroptosis is an iron-dependent form of non-apoptotic cell death and is associated with anti-cancer therapeutic efficacy. Further investigations are required to clarify the underlying mechanisms. Ferroptosis-resistant GC cell lines are constructed. Dysregulated mRNAs between ferroptosis-resistant and parental cell lines are identified. The expression of SOX13/SCAF1 is manipulated in GC cell lines where relevant biological and molecular analyses are performed. Molecular docking and computational screening are performed to screen potential inhibitors of SOX13. We show that SOX13 boosts protein remodeling of electron transport chain (ETC) complexes by directly transactivating SCAF1. This leads to increased supercomplexes (SCs) assembly, mitochondrial respiration, mitochondrial energetics and chemo- and immune-resistance. Zanamivir, reverts the ferroptosis-resistant phenotype via directly targeting SOX13 and promoting TRIM25-mediated ubiquitination and degradation of SOX13. Here we show, SOX13/SCAF1 are important in ferroptosis-resistance, and targeting SOX13 with zanamivir has therapeutic potential.

Gastric cancer (GC) ranks as the fifth most frequent cancer and the fourth leading cause of cancer-relevant death around the world[1,2]. In spite of enormous advances in GC management, patients with advanced GC still have a poor prognosis[3]. Cisplatin-based che-motherapies remain the first-line adjuvant or neoadjuvant therapy for advanced GC, but the acquaintance of chemotherapy resistance is still an obstacle to clinical efficacy[4]. Cancer immunotherapies, including immune checkpoint inhibitors, have emerged as effective therapies in various cancers[5]. However, only a small proportion of GC patients (approximately 15%) respond to immunotherapy due to resistance[6]. Hence, more research should be conducted to clarify the mechanism of chemo- and immuno-resistance.

Ferroptosis is considered a pervasive nonapoptotic cell death pathway culminating in excessive lipid peroxidation due to aberrant metabolism[7,8]. Ferroptosis has been linked to the efficacy of multiple anticancer therapies, including radiotherapy, chemotherapy, targeted

therapies and immunotherapy[9–12]. Erastin is a strong inhibitor of system Xc⁻ and RSL3 is the prototypical GPX4 inhibitor[7]. They work on different endogenous ferroptosis inhibitory systems; thus, we choose them for subsequent experiments. Metabolic reprogramming has been well recognized as one of the hallmarks of cancer and may be a therapeutically exploitable strategy to overcome drug resistance[13].

In this work, based on our previous research on cancer metabolism[10,14–16], we propose a potential metabolic mechanism that confers resistance to ferroptosis, chemotherapy and immunotherapy. We demonstrate that targeting SOX13/SCAF1 inhibits respiratory chain supercomplexes (SCs) assembly to overcome ferroptosis-mediated anticancer therapy resistance in gastric cancer.

## Results

### SOX13 dependence in ferroptosis-resistant GC cells

After analysis of the sensitivity of 20 tumor types to Erastin (a typical ferroptosis inducer) using the DepMap database, GC is the second most insensitive tumor type to Erastin (Supplementary Fig. 1A), suggesting that GC is relatively resistant to ferroptosis induction. Given that the role of ferroptosis has largely been unexplored, we generated Erastin-resistant (Erastin^resis) SNU-668 cells and RSL3-resistant (RSL3^resis) SNU-484 cells as described in Materials and Methods. They acquired resistance to treatment with Erastin or RSL3 (Supplementary Fig. 1B, C) but were still sensitive to inducers of apoptosis, including doxorubicin and gemcitabine (Supplementary Fig. 1D, E), compared to their parental cells in vitro. Erastin- or RSL3-induced lipid peroxidation (Supplementary Fig. 2A, B) and increased malondialdehyde (MDA) concentrations (Supplementary Fig. 2C) were noticeably attenuated in Erastin^resis SNU-668 cells and RSL3^resis SNU-484 cells compared to parental cells. Furthermore, RSL3^resis SNU-484 cells and Erastin^resis SNU-668 cells exhibited no significant differences in cell growth rate, apoptosis rate or cell cycle distribution from their parental cells (Supplementary Fig. 3A–C). Our data suggest that the acquired resistance to ferroptosis in GC cells is probably not attributed to disturbance of the cell cycle or apoptosis.

RNA sequencing (RNA-seq) analysis was performed in RSL3^resis SNU-484 cells and Erastin^resis SNU-668 cells compared to their parental cells. Furthermore, we examined the Cancer Therapeutics Response Portal (CTRP) datasets[17–19] and assessed associations between gene expression data across 654 cancer cell lines and cell sensitivities to ferroptosis inducers (FINs), including Erastin, RSL3, ML162 and ML210. Among the genes that were codysregulated during the acquisition of ferroptosis-resistance (absolute FC > 1.5, p < 0.05), we chose only genes that were strongly correlated with sensitivity to ferroptosis inducers in CTRP (with an absolute Pearson correlation Z-score >5) (Fig. 1A). Among these genes, we noticed that the transcript level of SOX13 was markedly higher in ferroptosis-resistant cells than in their isogenic parental cells (Fig. 1B) and was strongly correlated with ferroptosis-resistance in CTRP (Fig. 1C). Immunoblotting analysis confirmed the upregulation of SOX13 protein in resistant cells (Fig. 1D).

To determine the role of SOX13 in the ferroptosis-resistant phenotype of GC, we knocked down the SOX13 expression by short hairpin RNAs (Supplementary Fig. 4A, B) and then examined cell viability, lipid peroxidation and MDA concentration. RSL3-resistant (RSL3^resis) SNU-484 cells and Erastin-resistant (Erastin^resis) SNU-668 cells regained sensitivity to FINs with the knockdown of SOX13 (Fig. 1E–G, Supplementary Fig. 5A–C). This effect was wholly reversed by the ferroptosis inhibitor Fer-1 but not by the necroptosis inhibitor NSA or apoptosis inhibitor Z-VAD-FMK (Fig. 1E, 2A,B). The sensitization effect induced by SOX13 knockdown was reversed by co-transduction of a resistant SOX13 cDNA (SOX13^resis) (Supplementary Fig. 4A, B, Fig. 1E, Fig. 2A, B, Supplementary Fig. 5A–C). The association between high SOX13 expression and ferroptosis resistance were validated in 786-O (human clear cell renal cell carcinoma) cell line, which shows heightened sensitivity to ferroptosis inducers[20]. SOX13 over-expressing 786-O cells were less sensitive to

ferroptosis inducers, including Erastin and RSL3 (Supplementary Fig. 5D). According to NCCN guideline, cisplatin remains the first-line regimen for the treatment of locally advanced or metastatic gastric cancer[3]. A number of studies have indicated that ferroptosis is associated with the sensitivity to cisplatin[21–27]. We also found that RSL3-resistant (RSL3^resis) SNU-484 cells and Erastin-resistant (Erastin^resis) SNU-668 cells were resistant to cisplatin compared to their parental cells (Supplementary Fig. 1F). Knockdown of SOX13 increased the sensitivity to cisplatin in RSL3^resis SNU-484 and Erastin^resis SNU-668 cells, the sensitization effect induced by SOX13 knockdown was reversed by co-transduction of a resistant SOX13 cDNA (SOX13^resis) or ferroptosis inhibitor Fer-1 (Supplementary Fig. 5E, 6A). Cisplatin could indeed induce lipid peroxidation in RSL3^resis SNU-484 and Erastin^resis SNU-668 cells, and such an effect could be almost completely reversed by the ferroptosis inhibitor Fer-1 (Supplementary Fig. 6A). To facilitate the translation of the findings from Erastin or RSL3 resistance into clinic, we established the cisplatin-resistant (DDP^resis) SNU-668 cell line according to our previously reported procedures[28] for further experiments (Supplementary Fig. 6B). We found that the protein levels of SOX13/SCAF1 were higher in SNU-668 DDP^resis cell line compared to the parental cells. After the removal of the cisplatin stimulation for 7 days, we found that the expression of SOX13 remained stable (Supplementary Fig. 6C). Furthermore, we found that SNU-668 DDP^resis cell line acquired resistance to treatment with Erastin or RSL3 compared to the parental cells (Supplementary Fig. 6D). Knockdown of SOX13 increased the sensitivity to cisplatin, Erastin and RSL3 in SNU-668 DDP^resis cell line (Supplementary Fig. 6E). A number of studies have demonstrated that ferroptosis is involved in the resistance of the commonly used chemotherapeutic agents 5-FU and oxaliplatin[29–33]. We found that overexpression of SOX13 decreased the sensitivity to 5-FU and oxaliplatin in parental SNU-668 cells (Supplementary Fig. 6F).

### SOX13 overexpression in GC rendered resistance to different kinds of FINs in vivo

Exogenous expression of SOX13 in SNU-668 and SNU-484 cells (Supplementary Fig. 4C) markedly rendered the otherwise ferroptosis-sensitive cells resistance to FINs (Fig. 3A, B and Supplementary Fig. 7A, B). Xenograft assays illustrated that exogenous expression of SOX13 decreased the responsiveness to imidazole ketone Erastin (IKE, a metabolically stable analog of Erastin) and cisplatin compared with that of parallel negative control SNU-668 cells (SNU-668-NC) cells (Fig. 3C–E). The tumor growth inhibition treatment/control (T/C) ratios of IKE on tumors from SNU-668-NC cell-derived xenografts (CDX) and from SNU-668-SOX13-CDX tumors were 70.72 ± 13.81% and 31.88 ± 8.73%, respectively (Fig. 3F, left). The T/C ratios of cisplatin on SNU-668-NC-CDX and SNU-668-SOX13-CDX tumors were 63.82 ± 12.23% and 21.27 ± 6.19%, respectively (Fig. 3F, right). Treatment with IKE or cisplatin resulted in a higher level of PTGS2 (a well-known ferroptosis marker) in SNU-668-CDX tumors than in SNU-668-SOX13-CDX tumors (Fig. 3G). The effect of SOX13 on the sensitivity to IKE was validated in SNU-484 cells in vivo (Supplementary Fig. 8). We also found that continuous cisplatin treatment in vivo promoted the expression of SOX13 in SNU-668-CDX tumors (Fig. 3H), supporting the role of SOX13 in ferroptosis-resistance.

### SOX13 upregulated SCAF1 to boost ferroptosis-resistance

A previous study suggested that SOX13 promoted colorectal cancer migration, invasion, and metastasis by inducing epithelial-mesenchymal transition (EMT)[34], and the EMT process has been demonstrated to promote sensitivity to FINs[35,36]. However, we found that SOX13 induced no significant effects on the protein levels of Vimentin (a mesenchymal marker) or E-cadherin (an epithelial marker) (Supplementary Fig. 9), suggesting that the effect of SOX13 on EMT might be tumor specific.

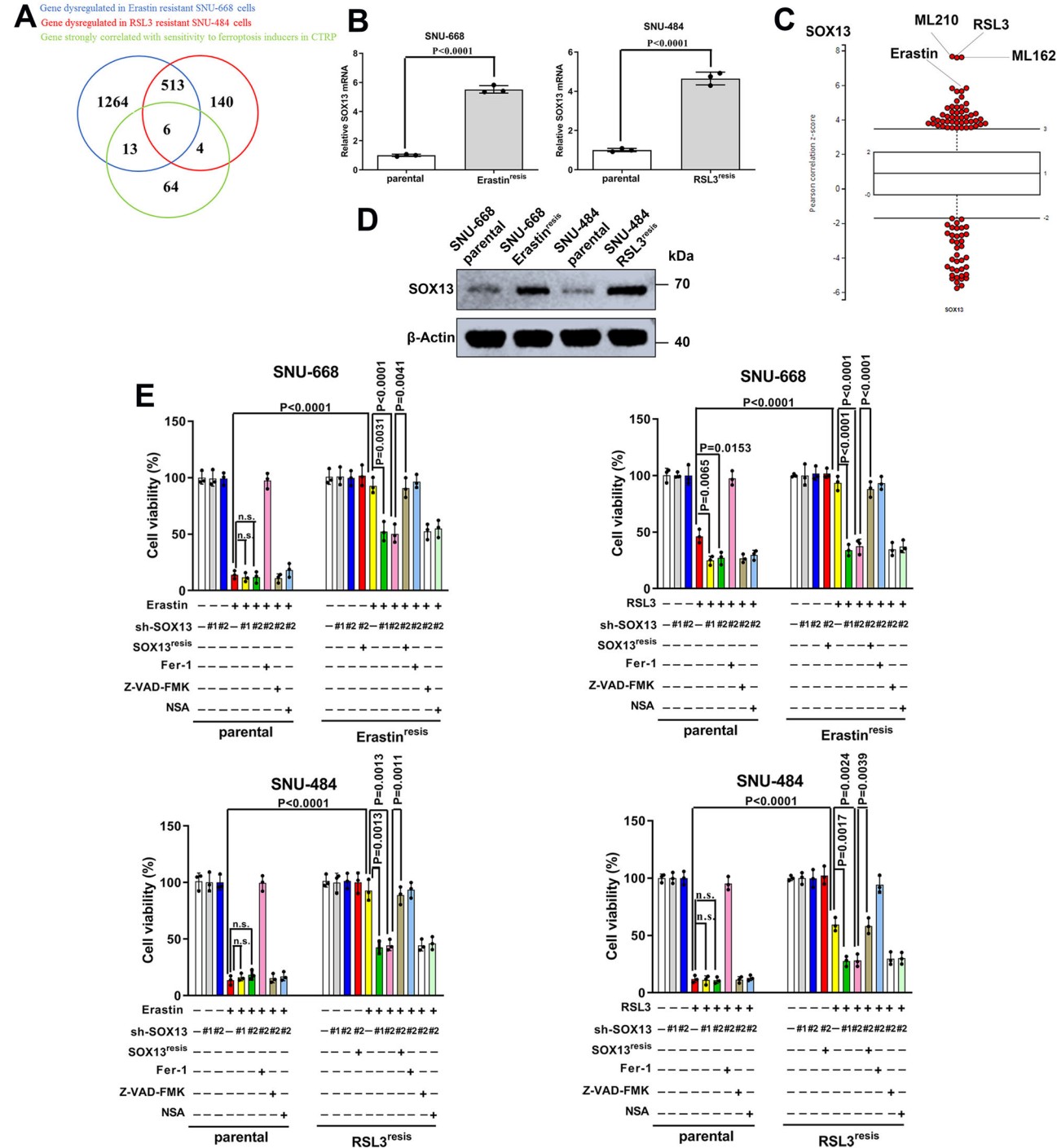

**Fig. 1 | SOX13 dependence in ferroptosis-resistant GC. A** Combinatorial analysis of genes differentially expressed between parental and ferroptosis-resistant cells and genes that are strongly correlated with sensitivity to ferroptosis inducers in CTRP. Six common genes were identified, among which was SOX13. **B** qRT–PCR analysis of SOX13 transcript levels in resistant cell lines ($n = 3$ independent experiments). RSL3$^{resis}$, RSL3 resistance; Erastin$^{resis}$, Erastin resistance. Data are presented as mean values ± SD. **C** Box-and-whisker plots show 5st and 95th percentile outlier compounds (red dots) where SOX13 expression levels are correlated with cell line sensitivity to the compounds in CTRP[17–19]. Plotted values are Z-scored Pearson's correlation coefficients. Line, median; box, 10th–90th percentiles. A positive z-score means that high expression is correlated with high resistance to compounds, and vice versa. **D** SOX13 was determined by immunoblotting assay and normalized to β-actin ($n = 3$ independent experiments). **E** Effect of SOX13 down-regulation and SOX13 re-expression on Erastin (2 μM) or RSL3 (0.5 μM) sensitivity in the absence or presence of Z-VAD-FMK (10 μM), NSA (1 μM), or Fer-1 (1 μM) in parental versus resistant cells. The cells were treated with FINs for 24 h ($n = 3$ independent experiments). Cell viability was evaluated with a CellTiter-Glo luminescent cell viability assay ($n = 3$ independent experiments). Data are presented as mean values ± SD. Statistical significance in (**B**) and (**E**) is determined by two-tailed unpaired *t* test. Source data are provided as a Source Data file.

To mechanistically study how the transcription factor SOX13 suppressed ferroptosis, we performed an RNA-sequencing analysis in SOX13-downregulated or control Erastin$^{resis}$ SNU-668 cells (Supplementary Fig. 10A). To identify the genes that are directly regulated by SOX13, we performed chromatin immunoprecipitation–sequencing (ChIP-seq) in the Erastin$^{resis}$ SNU-668 cells. Peak calling by the model-based analysis for the ChIP-seq algorithm revealed 21082 transcription sites bound directly to SOX13. When we compared the commonly

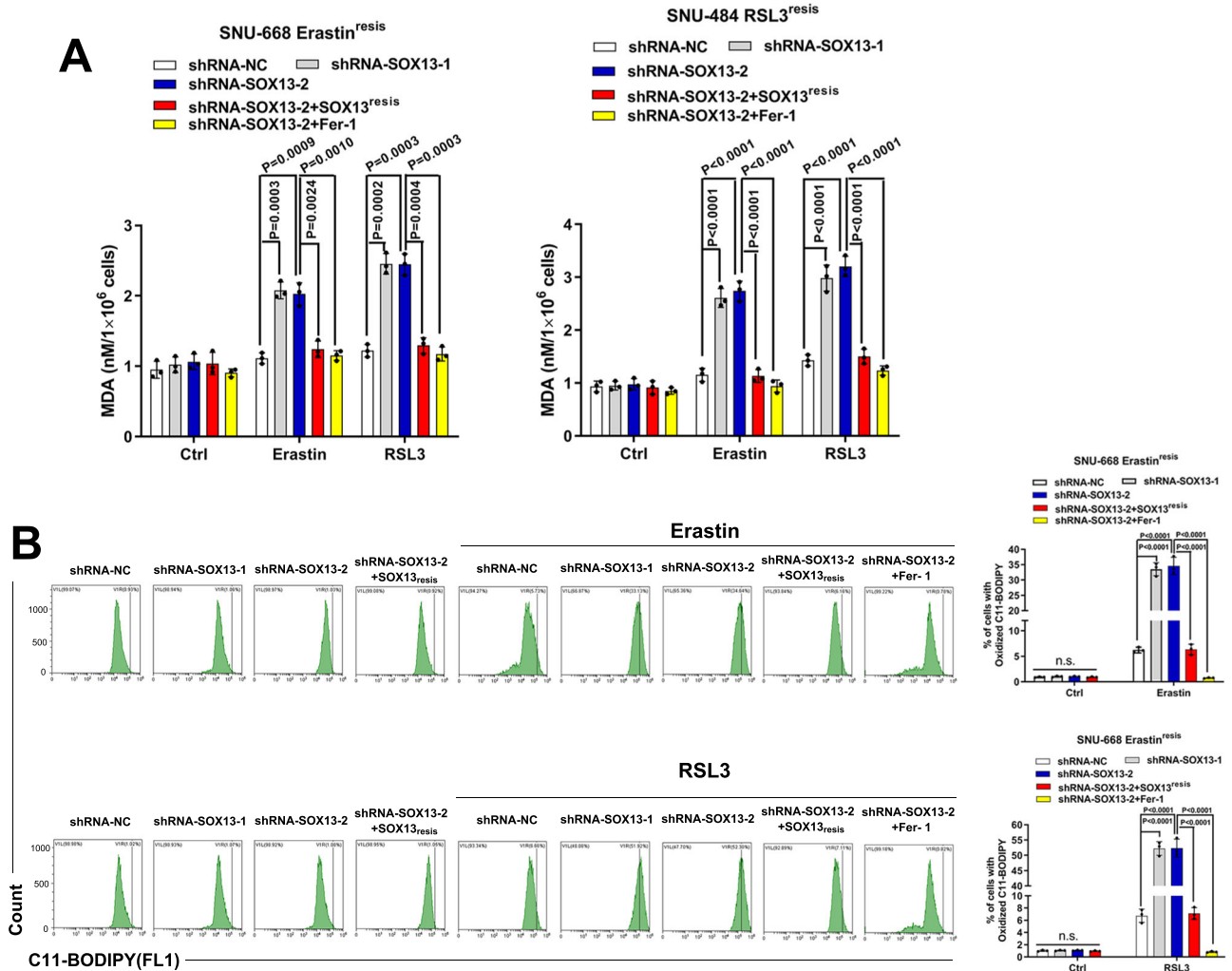

**Fig. 2 | SOX13 dependence in ferroptosis-resistant GC. A, B** Effect of SOX13 downregulation and SOX13 re-expression on Erastin (2 µM) or RSL3 (0.5 µM) sensitivity in the absence or presence of Fer-1 (1 µM) in resistant cells. The cells were treated with FINs for 24 h. **A** Intracellular MDA was assayed with ELISA ($n = 3$ independent experiments). Data are presented as mean values ± SD. **B** Lipid peroxidation was determined with a lipid peroxidation C11-BODIPY assay in SNU-668 Erastin^resis cells ($n = 3$ independent experiments), and a representative flow cytometry histogram plot is presented. Data are presented as mean values ± SD. Statistical significance in (**A**) and (**B**) is determined by two-tailed unpaired $t$ test. Source data are provided as a Source Data file.

dysregulated genes detected by RNA-seq with SOX13 downregulation, the genes with SOX13-bound transcription sites characterized by ChIP-seq and the genes dysregulated in FIN-resistant cells, we identified one overlapping gene SCAF1 (Fig. 4A). Untargeted metabolomic analysis of SOX13-depleted Erastin^resis SNU-668 cells indicated the metabolic profile change in SOX13-depleted cells (Supplementary Fig. 10B). The alanine, aspartate and glutamate metabolism, nicotinate and nicotinamide metabolism, glutathione metabolism and histidine metabolism ranked among the top differential pathways (Fig. 4B). We reanalyzed the genes in Fig. 4A. Of note, COX7A2L (SCAF1), a protein regulating the binding between CIII and CIV and increase NADH-dependent respiration[37,38], was the identified common gene. Knockdown of SCAF1 was reported to decrease the levels of NADPH, NADP and ATP[39], which ranked as the top dysregulated metabolites in SOX13-depleted Erastin^resis SNU-668 cells. Thus, we chose SCAF1 for subsequent experiments.

The transcript and protein levels of SCAF1 (Fig. 4C), but not its paralog COX7A2 (Supplementary Fig. 7C), were elevated in ferroptosis-resistant cell lines. The upregulation of SCAF1 was maintained by the elevation of SOX13, as demonstrated by decreased levels of SCAF1 with SOX13 silencing (Fig. 4D). Conversely, SOX13-overexpressing ferroptosis-sensitive cell lines exhibited higher levels of SCAF1 (Fig. 4E).

Binding profiles and peak calling records of SOX13 in the SCAF1 promoter shows that the 300-bp region upstream of the transcription start site of SCAF1 was highly enriched for SOX13 binding (Supplementary Fig. 10C). Then, we generated a dual-luciferase reporter construct containing the promoter region of SCAF1 and found that SCAF1 is highly responsive to SOX13-mediated transactivation (Fig. 4F). According to JASPAR, we identified three potential SOX13 binding motifs in the promoter area of SCAF1 (2000 bp upstream of the transcription start site) (Fig. 4G). Serial deletion and putative binding site-directed mutagenesis assays revealed that putative binding site 1 is vital for SOX13-dependent activation (Fig. 4G). ChIP−qPCR analysis further verified that SOX13 directly accumulates at SCAF1 promoter regions in GC cells with SOX13 upregulation and in FIN-resistant cells (Fig. 4H). In summary, the data indicate that SCAF1 is a direct transcriptional target of SOX13. The transcript level of SOX13 positively correlates with that of SCAF1 in multiple cancers, including GC, based on the databases TIMER (Supplementary Fig. 11) and GEPIA (Supplementary Fig. 12).

SCAF1 knockdown re-sensitized resistant cell lines to FINs and cisplatin, the sensitization effect induced by SCAF1 knockdown was reversed by co-transduction of a resistant SCAF1 cDNA (SCAF1^resis) (Fig. 5A, B and Supplementary Fig. 13A−D). SCAF1 knockdown also

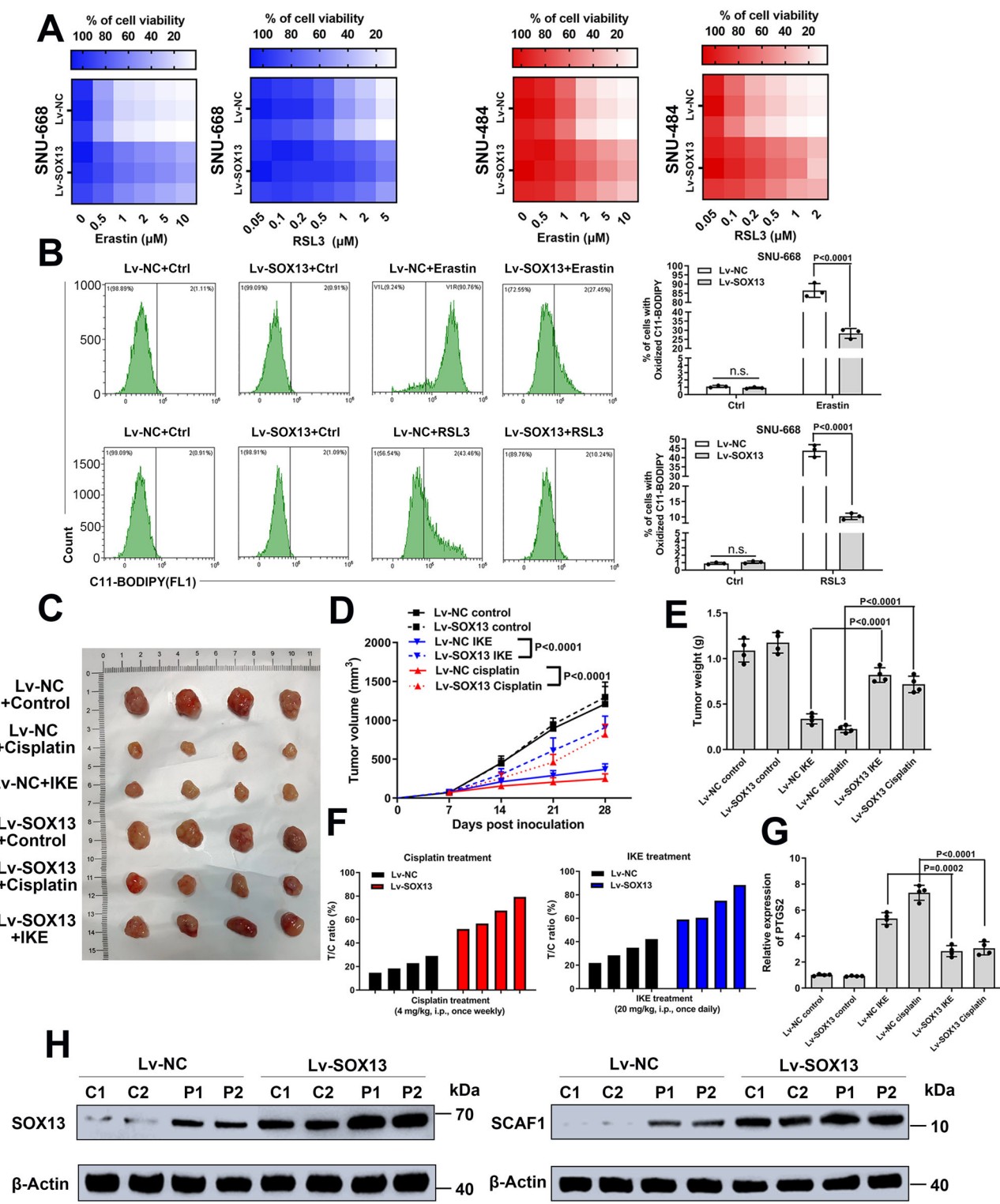

re-sensitized otherwise ferroptosis-resistant SOX13-overexpressing GC cells to FINs (Fig. 5C). The in vitro findings were further confirmed by in vivo xenograft assays. The SCAF1 gene was knocked out in SNU-668 cells with the CRISPR–Cas9 system. On Day 7 after injection, mice received IKE treatment. Tumors from SOX13-overexpressing xenografts were much less sensitive to IKE than tumors from control xenografts; however, such an effect was lost with SCAF1 knockout (Supplementary Fig. 14A–D). The transcript level of PTGS2 (a ferroptosis marker) was markedly increased with IKE treatment, and SOX13 overexpression attenuated that increase. Moreover, SCAF1 knockout

ameliorated the suppressive effect of exogenous SOX13 expression on PTGS2 levels. (Supplementary Fig. 14A–D)

In a study by ref. 40, they identified transcriptional alterations associated with acquired chemotherapy resistance from pre- and post-biopsy samples of the same patient with cisplatin and fluorouracil (CF) combination chemotherapy. We found that SOX13 and SCAF1 expression levels were significantly increased after acquiring resistance to CF combination chemotherapy (Supplementary Fig. 14E). These findings will further establish a link between ferroptosis resistance and clinical practice. In parallel with these findings, in an

**Fig. 3 | SOX13 is required and sufficient for ferroptosis-resistance. A, B** Effect of exogenous expression of SOX13 on sensitivity of parental SNU-668 cells (Left) or SNU-484 cells (Right) to the ferroptosis inducers RSL3 or Erastin. Cells were treated with lentivirus harboring the human SOX13 sequence or the empty vector, or in combination with various concentrations of Erastin (Left panel) or RSL3 (Right panel) for 24 h. The heatmap shows altered cell viability. **B** SNU-668 Cells were treated with lentivirus harboring the human SOX13 sequence or the empty vector, or in combination with Erastin (2 µM, Upper panel) or RSL3 (0.5 µM, Lower panel) for 24 h. Lipid peroxidation was determined using a lipid peroxidation C11-BODIPY assay ($n = 3$ independent experiments), and representative flow cytometry histogram plot is presented. Data are presented as mean values ± SD. (C-G) SNU-668-SOX13-CDX tumors and SNU-668-NC-CDX tumors were treated with cisplatin and IKE. Representative images of tumors formed (**C**), tumor growth curves (**D**), tumor weights (**E**), T/C ratio (**F**) and PTGS2 (**G**) expression analysis are shown. Treatment with IKE (20 mg/kg, i.p., once daily), cisplatin (4 mg/kg, i.p., once weekly) or PBS (100 µl, i.p., once daily) started on Day 7 and lasted for 3 consecutive weeks ($n = 4$ mice per group). T/C% = $T_{RTV}/C_{RTV} \times 100\%$; $T_{RTV}$ relative tumor volume after treatment; $C_{RTV}$ relative tumor volume of control group. Data are presented as mean values ± SD. **H** SOX13/SCAF1 expression in SNU-668 cell-derived xenografts was determined by immunoblotting and normalized to β-actin (Representative plot of experiments repeated in triplicate). C control, P cisplatin. Randomly selected two tumor samples per group are shown. Statistical significance in (**B**) and (**D, E, G**) is determined by two-tailed unpaired $t$ test. Source data are provided as a Source Data file.

independent cohort of patients who received cisplatin-based adjuvant chemotherapy ($n = 109$) (Supplementary Table 1)[10,41], we employed the immunohistochemistry auto-Stainer approach to assess SOX13/SCAF1 expression. The data revealed marked upregulation of SOX13/SCAF1 in GC samples compared to paired normal tissues (Supplementary Fig. 15A–C), consistent with TCGA data (Supplementary Fig. 15D). A significant correlation was found between the protein levels of SOX13 and SCAF1 in GC samples ($r = 0.406$, $p = 0.0007$, $p < 0.001$) but not in normal stomach samples ($r = 0.162$, $p = 0.2141$, $p > 0.05$) as evidenced by Pearson correlation analyses (Supplementary Table 5). SOX13/SCAF1 co-expression values were calculated by multiplying the immunohistochemical scores for these two molecules. GC patients were grouped according to the median coexpression value. Kaplan–Meier survival curves and log-rank test analyses illustrated that patients with high coexpression of SOX13/SCAF1 had lower disease-free survival (DFS) and overall survival (OS) rates (Fig. 6A). We found that the SOX13[high]SCAF1[high] is not associated with worse survival in patients receiving surgery only according to data from TCGA (Supplementary Fig. 15E). In another cohort of patients who received cisplatin-based neoadjuvant chemotherapy ($n = 52$) (Supplementary Table 2)[10], high coexpression of SOX13/SCAF1 was positively correlated with poor response to chemotherapy as evidenced by higher tumor regression grades (TRG) (Supplementary Table 6). To further consolidate the clinical relevance of ferroptosis to cisplatin-based chemotherapy in GC patients, we performed 4-hydroxy-2-nonenal (4-HNE, a well-known by-product of lipid peroxidation and is widely accepted as a stable marker for oxidative stress) staining in pre- and post-chemotherapy GC samples in cohort 2. Only weak 4-HNE staining was observed in pre-chemotherapy tumor samples; interestingly, cisplatin-based chemotherapy moderately or remarkably induced 4-HNE levels in GC samples (Fig. 6B). It suggests that cisplatin-based chemotherapy could indeed induce lipid peroxidation and therefore probably ferroptosis in GC patients. Then, we studied the correlation between 4-HNE levels in post-chemotherapy GC samples and patient outcomes. The analyses demonstrated that, strongly positive 4-HNE staining positively correlated with better chemotherapy response in GC patients (TRG ≤ 2) (Supplementary Table 7) and negatively correlated with high coexpression of SOX13/SCAF1 (Fig. 6C, Supplementary Table 8). In all, the results indicate that ferroptosis likely contributes to better chemotherapy response in GC patients.

## SOX13 boosts SCAF1-mediated assembly of respiratory chain supercomplexes and NADPH production

In light of the well-known role of SCAF1 in mitochondrial function, we set out to determine whether the ferroptosis-resistance phenotype was due to alterations in intrinsic mitochondrial function. An increase in the respiration of FIN-resistant GC cells was observed when compared to parental cells (Fig. 7A). Mitochondria were isolated from FIN-resistant GC cells or their parental cells, and both basal and state 3 respiration were determined. Complex I (CI) substrate-based cellular Oxygen Consumption Rate (OCR, driven by pyruvate and malate) and CI, combined CI+complex III (CIII), and complex IV (CIV) enzymatic

activity were elevated in FIN-resistant GC cells (Fig. 7B, C and Supplementary Fig. 16A, B). On the other hand, Complex II (CII) substrate-based cellular OCR (driven by succinate), CII activity, and combined CII + CIII activity did not differ between FINs resistant GC cells or their isogenic parental cells (Fig. 7B, C and Supplementary Fig. 16A, B). Obviously enhanced SC levels and activity (most notably SC I + III$_2$ + IVn) were observed (Fig. 7D and Supplementary Fig. 16C). These data are consistent with the specific increase in CI-driven respiration. Furthermore, SOX13 overexpression in SNU-668 and SNU-484 cells substantially increased SC levels and mitochondrial oxygen consumption; this effect was abolished with SCAF1 knockout (Fig. 7E, F and Supplementary Fig. 16D, E).

SCAF1, supercomplex assembly factor 1, has been reported to promote structural attachment between III$_2$ and IV and increase NADH-dependent respiration[35]. SCAF1-deficient cells exhibit decreased mitochondrial efficiency and NADH levels[38,42,43]. The effect of SOX13/SCAF1 on the production of NADH was confirmed in Supplementary Fig. 16F. SCAF1 has been reported to modulate the level of NADPH[39], and NADPH could boost the antioxidant defense in tumor cells by supporting the biosynthesis of GSH, Trx, and CoQ10-H2 to rescue them from ferroptosis[44–47]. As NADPH ranked among the top three dysregulated metabolites in SOX13-depleted RSL3[resis] SNU-668 cells and NADP(H) can be produced by phosphorylating NAD(H)[48,49], we explored whether SOX13/SCAF1 could regulate NADPH production through modulating NADH levels. FIN-resistant GC cells exhibited increased NADPH levels and NADPH/NADP$^+$ ratios compared with parental cell lines (Fig. 7H). Ectopic expression of SOX13 in sensitive cells upregulated the levels of NADPH, and this effect was abolished by SCAF1 knockout (Supplementary Fig. 16F). This finding suggests a dependence on SCAF1 for SOX13 mediated regulation of NADPH levels. To verify the role of SOX13/SCAF1 in production of NADPH was dependent on its effect on NADH production, we knocked down NAD kinases. NAD kinases (NADKs) are the only enzymes that generate NADP(H) by phosphorylating NAD(H)[48,49]. It has been reported that two kinds of NADKs that exist in human cells, NADK and NADK2, are mainly located in the cytosol and mitochondria, respectively[49]. We found that the cellular NADPH level was significantly decreased with silencing of the cytosolic enzyme NADK; however, downregulation of the mitochondrial enzyme NADK2 had no such effects (Supplementary Fig. 16G, H). Next, we explored which NAD kinases might participate in the maintenance of the ferroptosis-resistance phenotype. In parallel with this notion, the survival benefits of SOX13/SCAF1-overexpressing GC cells were largely eliminated with NADK silencing (Supplementary Fig. 16I). The in vivo data also provided evidence to the increased NADPH production when acquiring ferroptosis-resistance. Cisplatin treatment nonresponsive patients (TRG ≥ 3) exhibited increased NADPH abundance in blood compared with those who responded to cisplatin (TRG ≤ 2) from cohort 2 (Fig. 7I).

To further clarify the underlying mechanism of SOX13-mediated ferroptosis-resistance, we tested the effect of SOX13 on a series of well-known ferroptosis modulators. We found that suppression of SOX13 had no effect on the expression of GPX4, SLC7A11, or ACSL4 (Supplementary Fig. 17A), cellular divalent iron levels

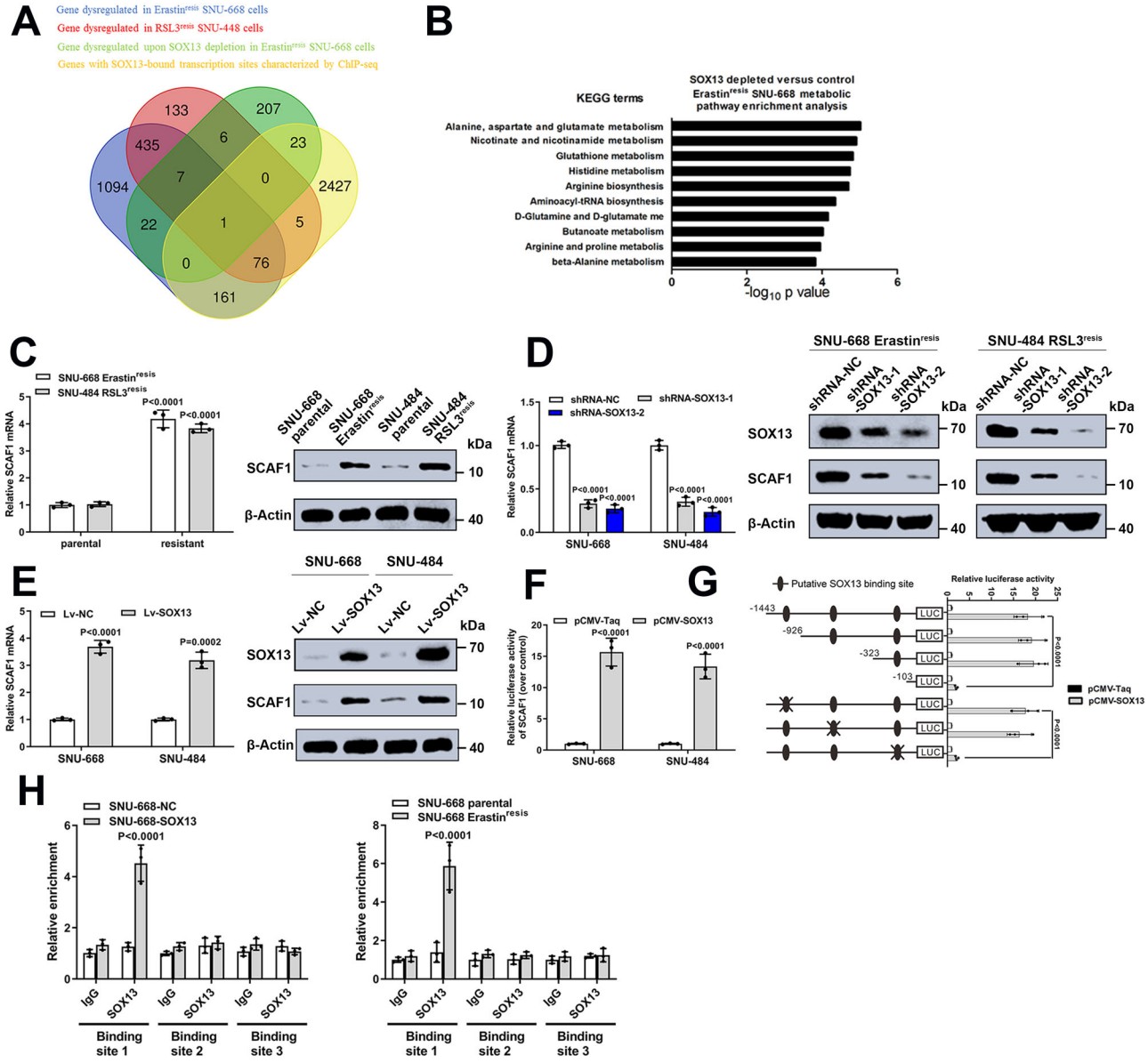

**Fig. 4 | SCAF1 is a direct transcriptional target of SOX13. A** Combinatorial analysis of genes differentially expressed between ferroptosis-sensitive and ferroptosis-resistant cells, genes dysregulated in response to SOX13 down-regulation in Erastin^resis SNU-668 cells, and the genes with SOX13-bound transcription sites characterized by ChIP-seq. The core common gene SCAF1 was identified. **B** Metabolite set enrichment analysis (MSEA) was used to determine the top ten metabolic pathways enriched in SOX13-downregulated Erastin^resis SNU-668 cells. **C** mRNA and protein levels of SCAF1 in resistant and parental SNU-668 and SNU-484 cell lines ($n = 3$ independent experiments). Data are presented as mean values ± SD. **D** Effect of SOX13 knockdown on SCAF1 protein abundance in resistant cells (Representative blot of three experimental replicates, quantification shows three independent experiments). Data are presented as mean values ± SD. **E** Effect of SOX13 overexpression on SCAF1 protein abundance in parental cells ($n = 3$ independent experiments). Data are presented as mean values ± SD. **F** Luciferase assay of GC cells cotransfected with firefly luciferase constructs containing the SCAF1 promoter and pCMV-SOX13 ($n = 3$ independent experiments). Data are presented as mean values ± SD. **G** Luciferase assay of GC cells cotransfected with firefly luciferase constructs containing a series of SOX13 promoter deletion mutants and pCMV-SOX13 ($n = 3$ independent experiments). Data are presented as mean values ± SD. **H** ChIP-qPCR analysis further verified that SOX13 directly accumulates at SCAF1 promoter regions in GC cells with SOX13 upregulation and in Erastin^resis SNU-668 cells ($n = 3$ independent experiments). Data are presented as mean values ± SD. Statistical significance in (**B**–**H**) is determined by two-tailed unpaired $t$-test. Source data are provided as a Source Data file.

(Supplementary Fig. 17B), system Xc⁻ (Supplementary Fig. 17A) and GPX4 (Supplementary Fig. 17D) activity or cellular phospholipid composition (Supplementary Fig. 17E).

### Targeting the SOX13–SCAF1 pathway potentiates the antitumor activity of immunotherapy

Cancer immunotherapy has been recently noted for its association with ferroptotic cell death in vivo[9,11,12,50]. Next, we examined whether combining anti−PD-1 treatment with targeting of SOX13/SCAF1 would

enhance the antitumor efficacy in vivo. Whereas PD1 antibody-targeted monotherapy moderately inhibited the growth of tumors from wild-type YTH16 (m) xenografts, anti-PD1 antibody monotherapy caused a more pronounced suppressive effect on tumor growth in tumors from YTH16 (m) cells with SOX13 knockout. In addition, the ferroptosis inhibitor Lipro-1 notably attenuated the antitumour effect mediated by PD1 and SOX13 knockdown. Exogenous expression of SCAF1 abrogated this suppressive effect of anti-PD1 antibody monotherapy. However, no significant differences existed in growth between

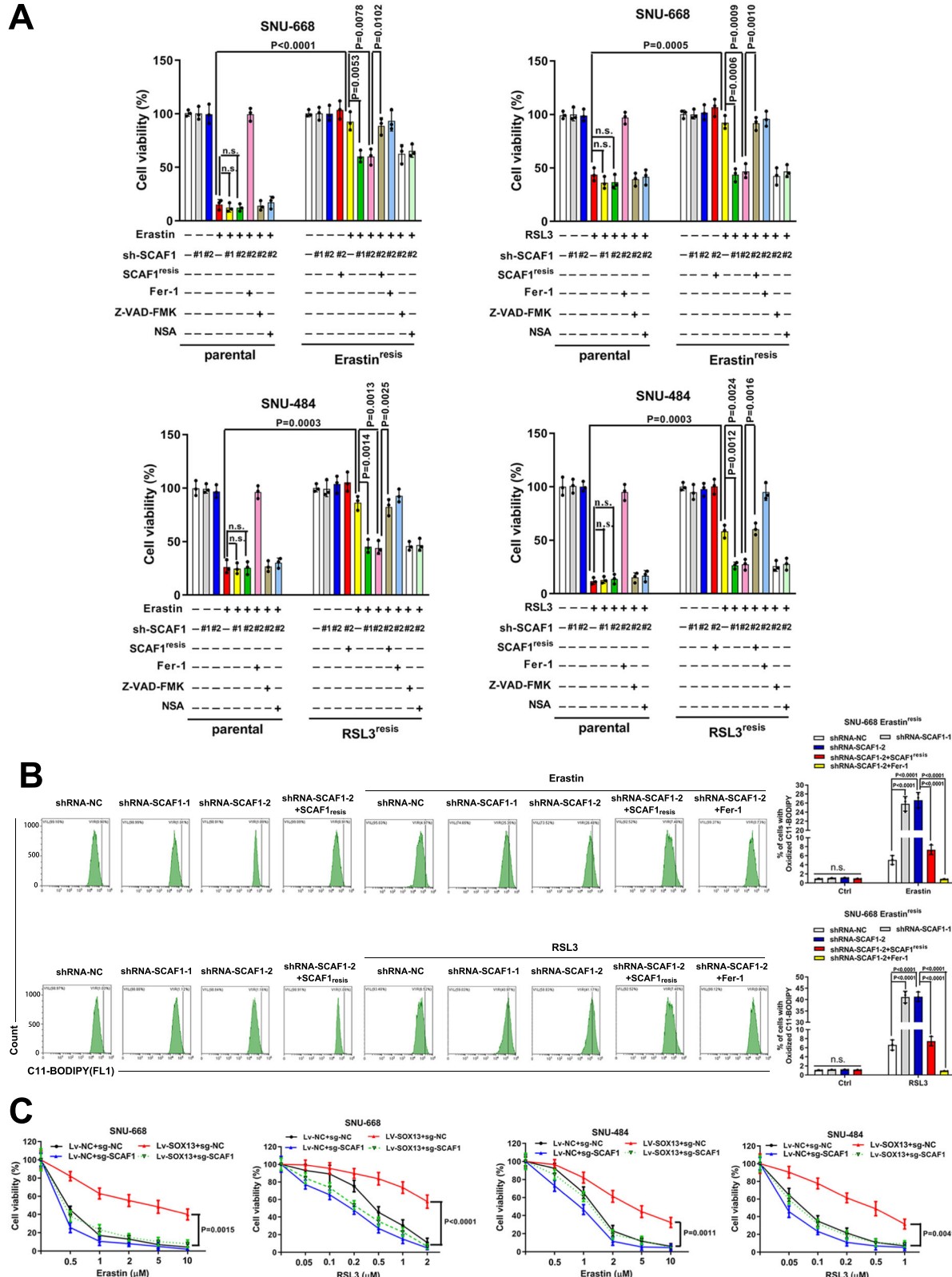

**Fig. 5 | SOX13 upregulated SCAF1 to boost ferroptosis-resistance. A** Effect of SCAF1 knockdown and SCAF1 re-expression on Erastin (2 μM) or RSL3 (0.5 μM) sensitivity in the absence or presence of Z-VAD-FMK (10 μM), NSA (1 μM), or Fer-1 (1 μM) in parental versus resistant cells. The cells were treated for 24 h. Cell viability was evaluated with a CellTiter-Glo luminescent cell viability assay (*n* = 3 independent experiments). Data are presented as mean values ± SD. **B** SNU-668 Erastin[resis] cells were treated with lentiviruses encoding SCAF1 shRNA-1 or shRNA-2 or a scrambled shRNA alone and/or resistant SOX13 cDNA (SOX13[resis]), or in combination with Erastin (2 μM) or RSL3 (0.5 μM) for 24 h. Lipid peroxidation was determined with a lipid peroxidation C11-BODIPY assay in SNU-668 Erastin[resis] cells (*n* = 3 independent experiments), and a representative flow cytometry histogram plot is presented. Data are presented as mean values ± SD. **C** Effect of SOX13 overexpression (Lv-SOX13) alone, SCAF1 KO (sg-SCAF1) alone, or the two in combination on Erastin or RSL3 sensitivity in parental SNU-668 and SNU-484 cells. The cells were treated for 24 h (*n* = 3 independent experiments). Data are presented as mean values ± SD. Statistical significance in (**A**–**C**) is determined by two-tailed unpaired *t* test. Source data are provided as a Source Data file.

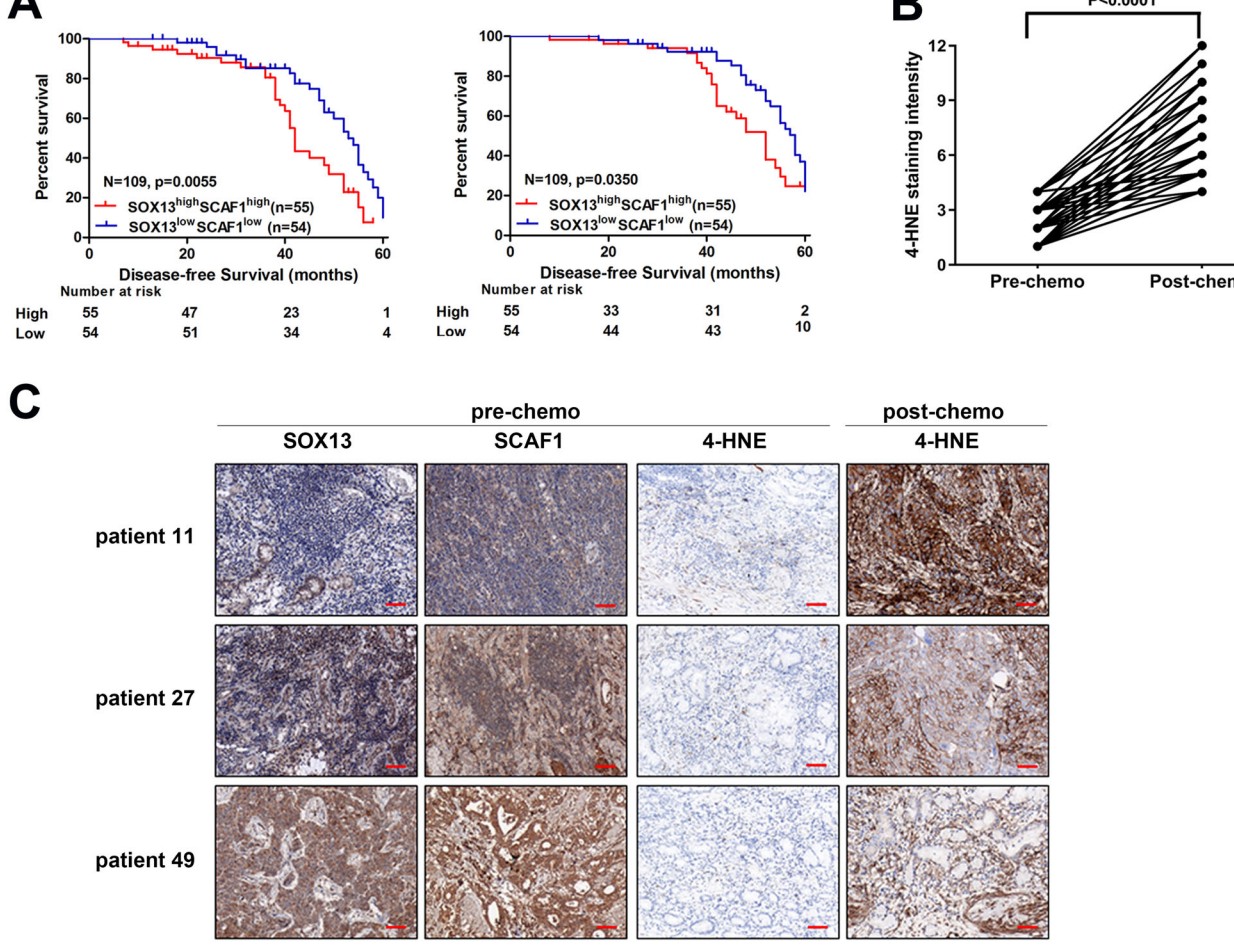

**Fig. 6 | Ferroptosis induction correlates with GC patient response to chemotherapy. A** Kaplan–Meier survival curves and log-rank test analysis of DFS (left) and OS (right) rates in GC cancer patients who underwent cisplatin-based adjuvant chemotherapy. The GC patients were grouped according to the median coexpression value within their tumors. **B** Immunochemistry scoring of 4-hydroxy-2-nonenal (4-HNE) staining of matched GC samples before and after cisplatin-based chemotherapy from 52 GC patients. Error bars are means ± SD, $n$ = 3 randomly selected magnification fields. **C** Representative images of 4-HNE, SOX13 and SCAF1 immunohistochemical staining of matched GC samples from the same patients before or after chemotherapy. Scale bar: 50 μm. Two-tailed paired $t$ test (**B**) or two-sided log-rank test (**A**). Source data are provided as a Source Data file.

untreated wild-type, SCAF1-overexpressing and SOX13-knockout YTH16 (m) tumors. (Fig. 8A–C) We also detected an increased proportion of cytotoxic CD8⁺ T and CD4⁺ T cells as well as a marked increase in lipid peroxidation in CD45⁻ cells in immunocompetent C57BL/6 mice bearing YTH16 (m)-SOX13-knockout xenograft tumors but not in mice bearing YTH16 (m)-SCAF1-overexpression xenograft tumors following anti-PD1 treatment. The ferroptosis inhibitor Lipro-1 markedly ameliorated such an effect mediated by PD1 and SOX13 knockdown. (Fig. 8D, E, Supplementary Fig. 18) The data suggest that activation of the SOX13–SCAF1 signal axis diminished the immunotherapeutic efficacy via inhibiting ferroptosis.

## Zanamivir directly targets SOX13 to suppress ferroptosis-resistance in GC

Given the vital role of SOX13 in ferroptosis-resistance, a molecular docking analysis with SOX13 was performed using more than 2000,000 different compounds. The DNA binding fragment (position 424-492), which was validated to be important for its role as a transcriptional factor[51], was defined as the screening target site. As a result, 20 compounds with the highest scores were identified as putative ligands and selected for further functional screenings (Fig. 9A, Supplementary Table 9). Among the identified compounds, T2529, T0278, T4706 and T2720 most significantly increased FINs induced growth

inhibition in ferroptosis-resistant cell lines (Fig. 9B). We further purified the SOX13 protein (Supplementary Fig. 19A) and examined the interaction between SOX13 and five screened compounds by SPR. Zanamivir (T2529 in the compound library), the first neuraminidase inhibitor approved for the prevention and treatment of influenza, had the lowest $K_D$ values ($6.87 \times 10^{-6}$) for SOX13 protein (Fig. 9C, Supplementary Fig. 19B). Computational molecular docking analysis showed that the potential compound zanamivir binds to the functional pocket of SOX13 protein (Supplementary Fig. 19C, D). We further showed that only zanamivir inhibited SOX13 protein level (Supplementary Fig. 19E). Subsequently, a Cellular Thermal Shift Assay (CETSA)[52,53] was performed to assess zanamivir target engagement and an inhibitory effect of zanamivir on the degradation rate of SOX13 with increase of temperature was observed (Supplementary Fig. 19F). It suggests that zanamivir can physically bind to SOX13. Furthermore, zanamivir significantly made the ferroptosis-resistant cells more sensitive to FINs dose-dependently (Fig. 9D, Supplementary Fig. 20A, B) without significant effects on cell proliferation, cell cycle and apoptosis (Supplementary Fig. 21A–C).

We set out to assess the role of SOX13 in the biological function of zanamivir. The data of lipid peroxidation assay revealed that re-overexpression of SCAF1 in SOX13-knockdown GC cells, but not re-overexpression of SOX13, abolished the increased sensitivity to FINs

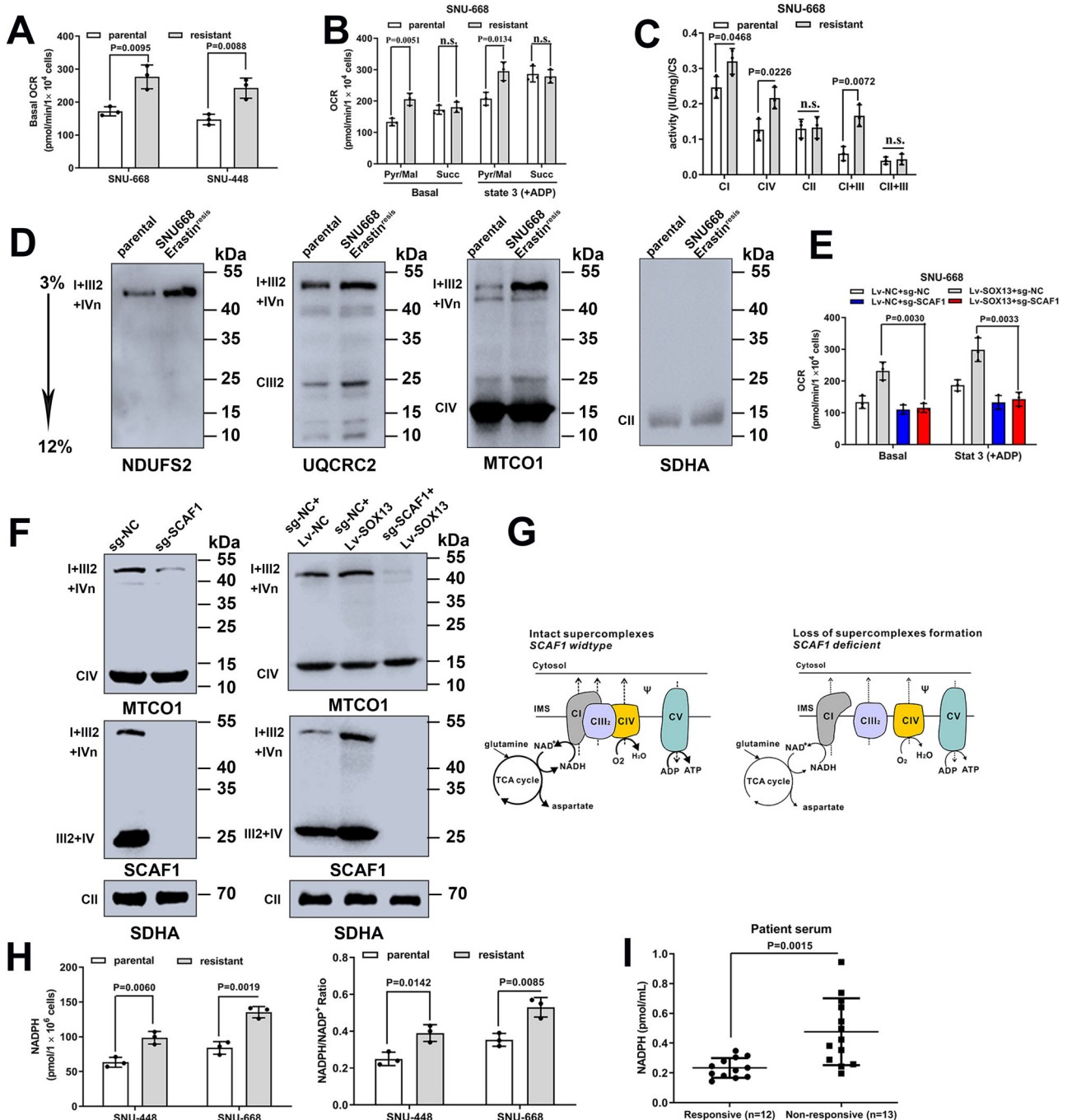

**Fig. 7 | SOX13 boosted SCAF1-mediated assembly of respiratory chain supercomplexes and NADPH production.** Parental and resistant cells. Oxygen consumption analysis was performed in (**A**) intact cells or (**B**) isolated mitochondria using pyruvate-malate or succinate as specific substrates, respectively ($n = 3$ independent experiments). Data are presented as mean values ± SD. **C** Mitochondrial enzymatic activities of complex I (CI), complex IV (CIV), complex II (CII), CI+complex III (CIII), and CII + III normalized to citrate synthase (CS) levels in parental and resistant SNU-668 cells ($n = 3$ independent experiments). Data are presented as mean values ± SD. **D** Western blot analysis of the indicated proteins, including CI, CIII, CIV, and CII, after blue native PAGE (BN-PAGE) of digitonin-solubilized mitochondria from parental and resistant SNU-668 cells (Representative plot of experiments repeated in triplicate). **E** Oxygen consumption rates (OCR) of isolated mitochondria from parental SNU-668 cells with SOX13 overexpression (Lv-SOX13)

alone, SCAF1 KO (sg-SCAF1) alone or the two in combination ($n = 3$ independent experiments). Data are presented as mean values ± SD. **F** SC levels in parental SNU-668 cells with SOX13 overexpression (Lv-SOX13) alone, SCAF1 KO (sg-SCAF1) alone or the two in combination (Representative plot of experiments repeated in triplicate). **G** Schematic illustrating the metabolic consequences from loss of supercomplex formation in GC cells. Intact supercomplexes (left) and loss of supercomplexes (right) are demonstrated using the respirasome (complexes I, III₂, and IV). **H** NADPH levels and NADPH/NADP+ in parental and resistant cells ($n = 3$ independent experiments). Data are presented as mean values ± SD. **I** Abundance of NADPH in the blood of cisplatin treatment nonresponsive patients (TRG ≥ 3, $n = 13$) compared to patients sensitive to cisplatin (TRG ≤ 2, $n = 12$) from cohort 2. Data are presented as mean values ± SD. Two-tailed unpaired $t$-test (**A**–**C**, **E**, **H**) or two-tailed Mann–Whitney test (**I**). Source data are provided as a Source Data file.

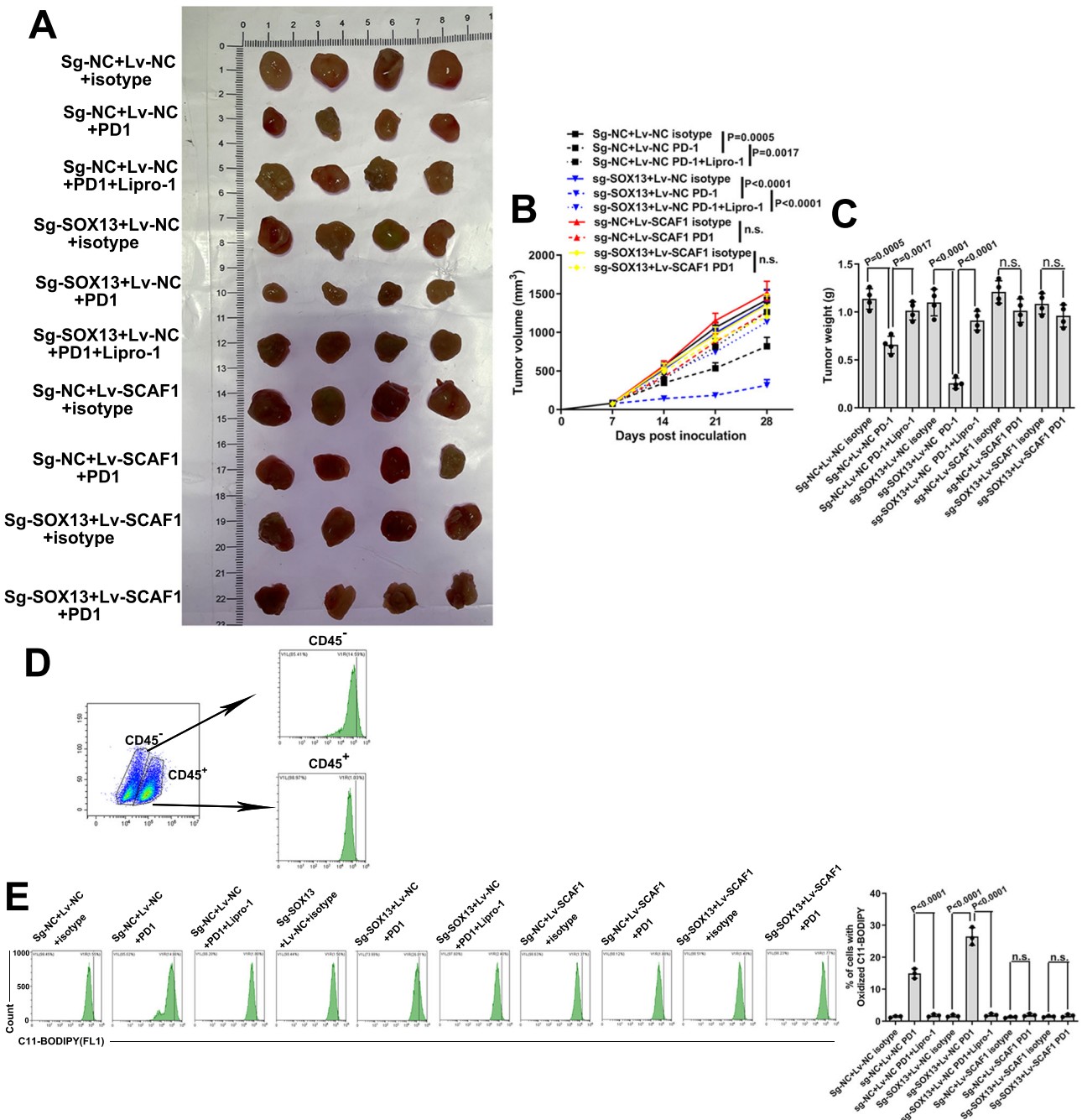

**Fig. 8 | Targeting the SOX13–SCAF1 pathway potentiates the antitumor activity of immunotherapy. A** SOX13-KO (sg-SOX13), SCAF1-overexpressing (Lv-SCAF1) or control YTH16 (m) cells were grown as xenografts. Antibody to mouse PD1 treatment or lipoxstatin alone or in combination started on Day 7 after injection of C57BL/6 mice with cells to form xenografts. Tumor growth (**B**) and weight change (**C**) are shown (*n* = 4 mice per group). Data are presented as mean values ± SD. **D** Flow cytometry analysis of BODIPY fluorescence in CD45+ and CD45− tumor cells isolated from control YTH16 (m) derived xenografts treated with antibody to mouse PD1. **E** Representative flow cytometry analysis and quantification of BODIPY fluorescence in CD45− tumor cells from YTH16 (m)-derived xenografts with the treatment described above (data from randomly selected three tumors from each group). Data are presented as mean values ± SD. Statistical significance in (B,C,E) is determined by two-tailed unpaired *t*-test. Source data are provided as a Source Data file.

in presence of zanamivir (Supplementary Fig. 22). Such a phenomenon was validated in an extended cell line MKN45 (Supplementary Fig. 23). Next, we evaluated the part that SOX13 played in the ferroptosis-sensitizing activity of zanamivir in vivo. The synergistic antitumor effect of zanamivir and IKE was attenuated with SOX13-knockdown, however, such an effect was recovered with re-overexpression of wild-type SOX13 (Fig. 9E–G). In general, we show that zanamivir directly binds to SOX13, suppresses SOX13 expression and SOX13-mediated ferroptosis-resistance. Subsequently, we

verified the limited toxicity of zanamivir in vivo. No marked signs of toxicity, such as loss of body weight (Supplementary Fig. 24A), loss of appetite, and decreased activity during treatment were observed. In addition, zanamivir treatment induced no significant change in size and cell morphology of selected organs (Supplementary Fig. 24B, C). No significant cytotoxicity of zanamivir was found on human pulmonary epithelial cells, human umbilical vein endothelial cells and on nontumorigenic mammary cells at the tested concentrations (Supplementary Fig. 25A, B).

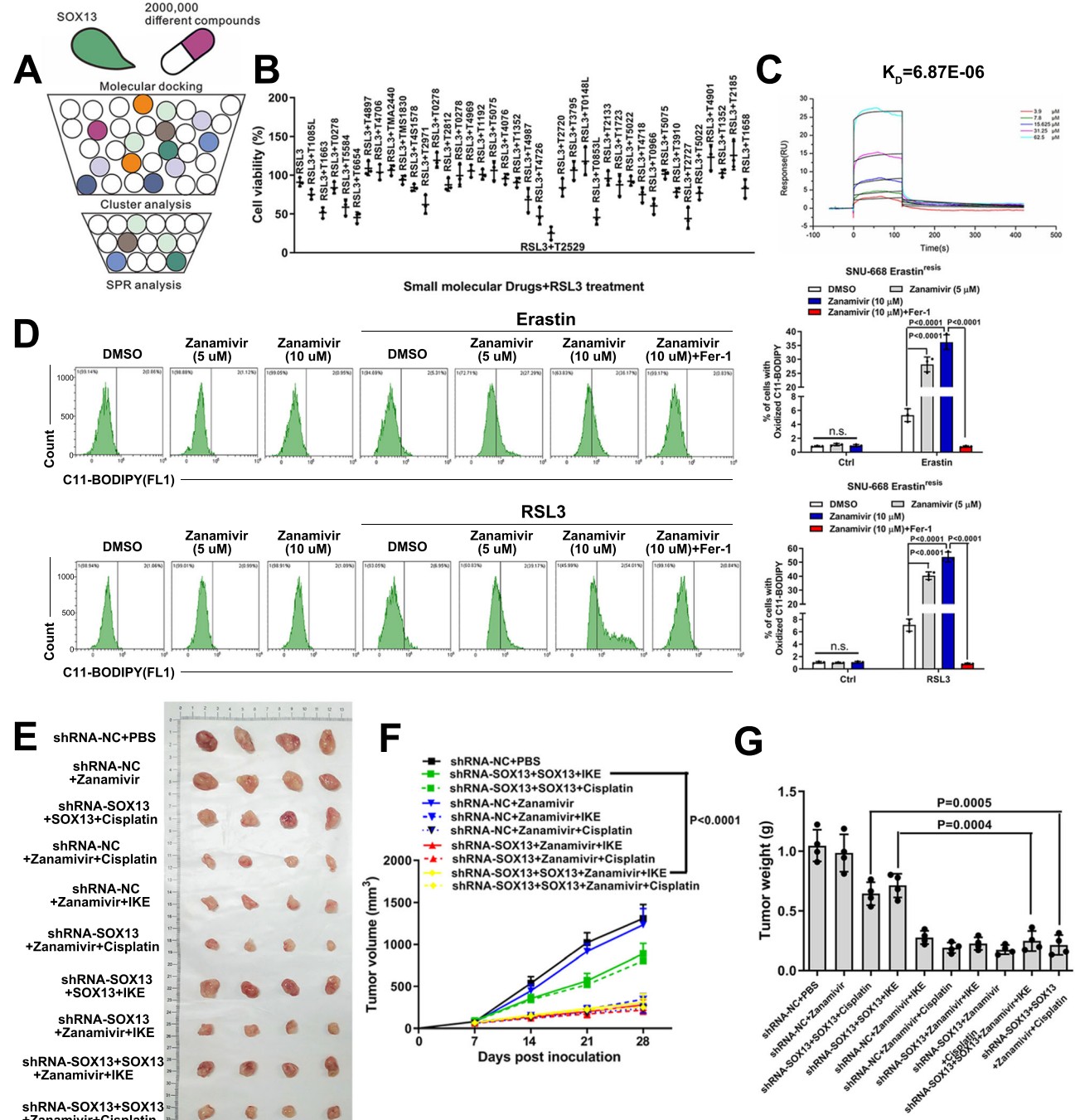

**Fig. 9 | Zanamivir directly targets SOX13 to inhibit ferroptosis-resistance in GC.**
**A** A schematic diagram demonstrating the screening strategy for SOX13-targeting compounds. **B** Effect of the top 40 candidate compounds (10 μM) on RSL3 (0.5 μM) sensitivity in Erastin[resis] SNU-668 cells. The cells were pretreated with candidate compounds (10 μM) for 12 h prior to being exposed to RSL3 (0.5 μM) for 24 h. Cell viability was examined ($n = 3$ independent experiments). Data are presented as mean values ± SD. **C** SPR analysis of the interaction between zanamivir and SOX13 protein. **D** The Erastin[resis] SNU-668 cells cells were pretreated with zanamivir (0 μM, 5 μM, 10 μM) for 12 h and then exposed to Erastin (2 μM) or RSL3 (0.5 μM) for 24 h. Lipid peroxidation was determined using a lipid peroxidation C11-BODIPY assay

($n = 3$ independent experiments), and representative flow cytometry histogram plot is presented. Data are presented as mean values ± SD. Tumors from Erastin[resis] SNU-668 cells transfected with desired vector were treated with zanamivir, cisplatin and IKE. Representative images of tumors formed (**E**), tumor growth curves (**F**) and tumor weights (**G**) are shown. Treatment with cisplatin (4 mg/kg, i.p., once weekly), IKE (20 mg/kg, i.p., once daily), zanamivir (5 mg/kg, i.p., once daily) or PBS (100 μl, i.p., once daily) started on Day 7 and lasted for 3 consecutive weeks ($n = 4$ mice per group). Data are presented as mean values ± SD. Statistical significance in (**D**, **F**, **G**) is determined by two-tailed unpaired *t* test. Source data are provided as a Source Data file.

## The E3 ligase TRIM25 modulates zanamivir-induced poly-ubiquitination and proteasomal degradation of SOX13

As illustrated in Fig. 10A and Supplementary Fig. 26A, zanamivir treatment induced marked decreased SOX13 and the SOX13 target gene SCAF1 protein abundances in a dose-dependent manner, which

hints that zanamivir might induce SOX13 degradation. Moreover, the protein levels of SOX2, SOX4, and SOX9 were not significantly altered upon zanamivir treatment (Supplementary Fig. 26B), which confirmed the selectivity of zanamivir. Next, a cycloheximide (CHX) chase assay was performed to examine the effect of zanamivir on

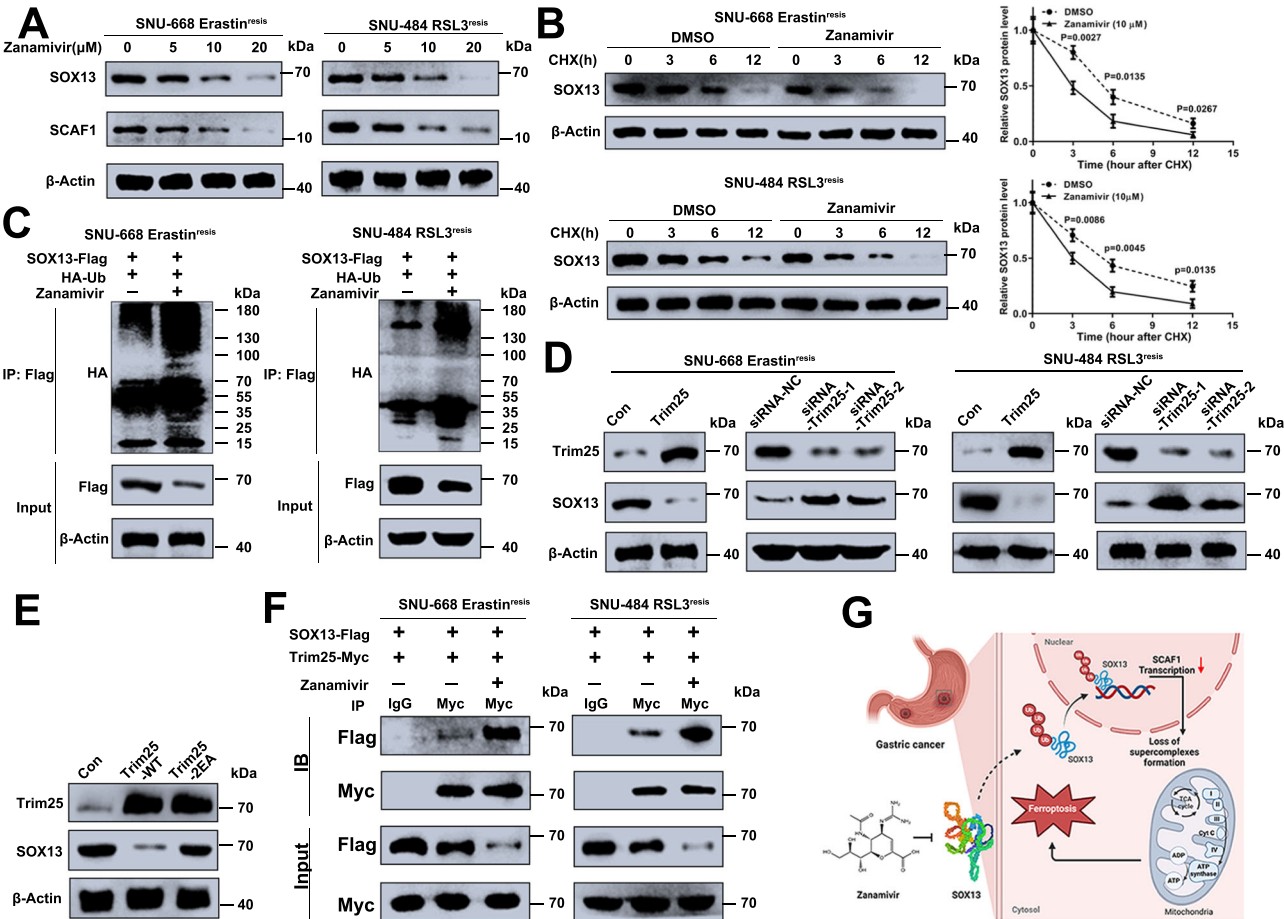

**Fig. 10 | The E3 ligase TRIM25 modulates zanamivir-induced polyubiquitination and proteasomal degradation of SOX13. A** The Erastin[resis] SNU-668 cells or RSL3[resis] SNU-484 cells were pretreated with zanamivir (0 μM, 5 μM, 10 μM, 20 μM) for 36 h. Western blot analysis of SOX13, and SCAF1 protein expression levels (Representative plot of experiments repeated in triplicate). **B** The Erastin[resis] SNU-668 cells or RSL3[resis] SNU-484 cells were treated with cycloheximide (100 μg/mL) as indicated in the presence or absence of zanamivir (10 μM) (n = 3 independent experiments). Data are presented as mean values ± SD. **C** The SNU-668 cells or SNU-484 cells were transfected with the desired plasmids, 36 h after transfection, they were treated with zanamivir (10 μM) for 24 h, and then subjected to IP using anti-Flag antibody followed by Western blot analysis (Representative plot of experiments repeated in triplicate). **D** Depletion of TRIM25 increased while forced expression of TRIM25 decreased the protein level of SOX13 in Erastin[resis] SNU-668 cells or RSL3[resis] SNU-484 cells (Representative plot of experiments repeated in triplicate). **E** Western blot analysis of SOX13 and TRIM25 protein levels in Erastin[resis] SNU-668 cells overexpressing TRIM25-WT, TRIM25-2EA or vector control (Representative plot of experiments repeated in triplicate). **F** Co-IP assay was used to determine the binding of SOX13 with TRIM25 in the presence or absence of zanamivir (Representative plot of experiments repeated in triplicate). **G** Ferroptosis-sensitization mechanism of SOX13-targeting compound zanamivir. Zanamivir targets SOX13 protein and thus downregulates SCAF1, which leads to decreased supercomplexes (SCs) assembly, mitochondrial respiration, mitochondrial energetics, and increased ferroptosis sensitivity. Statistical significance in (**B**) is determined by two-tailed unpaired t test. Source data are provided as a Source Data file.

SOX13 stability. We found that zanamivir induced an obvious increase in degradation of SOX13 (Fig. 10B). Accordingly, a robust increase in SOX13 polyubiquitination was observed in FIN-resistant GC cells upon zanamivir treatment (Fig. 10C). Taken together, zanamivir promotes the ubiquitination and degradation of SOX13. To illustrate the potential regulator of SOX13, we next explored the potential ubiquitin ligases[54] that bind to SOX13 protein and promote its proteasomal degradation by searching a protein-protein interaction database BioGRID (https://thebiogrid.org/). Among the top three ubiquitin ligases predicted to bind to SOX13, ectopic expression of TRIM25 rather than FBXO28 or TRAF2 downregulated the protein level of SOX13 (Supplementary Fig. 26C). Consistently, depletion of TRIM25 increased while forced expression of TRIM25 decreased the protein level of SOX13 (Fig. 10D). The mRNA level of SOX13 was not changed by TRIM25 manipulation (Supplementary Fig. 26D). These results indicate that TRIM25 probably modulates SOX13 degradation. In line with this, exogenous expression the wild-type TRIM25, other than TRIM25-2EA mutant lacking E3 ligase activity[55] resulted in a noticeable decrease in SOX13 expression

(Fig. 10E). Moreover, we showed that the E3 ligase activity of TRIM25 was indispensable for reversing the ferroptosis-resistant phenotype as TRIM25-2EA mutant failed to sensitize FIN-resistant cell lines (Supplementary Fig. 26E).

Furthermore, exogenous SOX13 co-immunoprecipitated with TRIM25 (Supplementary Fig. 26F). The interaction between endogenous SOX13 and TRIM25 was enhanced with zanamivir treatment (Fig. 10F). To conclude, these findings demonstrate that zanamivir directly binds to SOX13 to increase TRIM25-mediated SOX13 protein destabilization, thus inactivating the SOX13-SCAF1 signaling axis to inhibit ferroptosis-resistance.

## Discussion

Metabolic rewiring has been shown to facilitate cancer initiation and progression[13]. This notion has constantly inspired our research team to develop strategies targeting key molecules in cancer metabolism reprogramming for cancer therapeutics[10,14–16]. The link between metabolic rewiring and drug resistance has been well recognized[13]. Among the molecular mechanisms of drug resistance, ferroptosis is

considered to be the coalescence of metabolic perturbation and dysfunction of the main ferroptosis surveillance systems[7].

In this study, we cast light on a potential metabolic mechanism that confers resistance to ferroptosis and ferroptosis-mediated therapeutics, including cisplatin and immunotherapy in GC. We showed that SOX13 boosts protein remodeling of electron transport chain (ETC) complexes by transactivating the expression of SCAF1. This leads to increased SCs assembly, mitochondrial respiration, mitochondrial energetics and drug resistance. Inhibition of SOX13/SCAF1 restored sensitivity in ferroptosis-resistant GC cell lines and reversed chemo- and immune resistance in CDX mouse models.

SOX13 is a member of the SRY-related high motility group (HMG) box (SOX) family of transcription factor. Research has focused on other members of the SOX family in carcinogenesis[56,57], and the role of SOX13 has been neglected. Du et al. reported that the upregulation of SOX13 enhanced colorectal cancer metastasis by inducing the EMT process[34]. Nevertheless, the role of SOX13 in treatment resistance, especially molecularly targeted therapy, remains elusive. We revealed that SOX13 promotes ferroptosis resistance-mediated therapy resistance in GC. Ferroptosis-mediated cell death contributes to chemo- and immune- therapeutic efficacy, and it is highly feasible to discover potential therapeutic targets and monitoring biomarkers for chemotherapy by illuminating the vital modulators of ferroptosis.

The mitochondrial energetic metabolism and ferroptosis closely interact with each other[58]. However, there are still some controversies concerning the role of energetic metabolism in ferroptotic process[59–62]. In this study, we demonstrated that SOX13 boosts NADPH production by upregulating SCAF1 to drive ferroptosis-resistance. SCAF1, supercomplex assembly factor 1, has been reported to promote structural attachment between $III_2$ and IV and increase NADH-dependent respiration[37,38]. SCAF1-knockout zebrafish displayed a smaller size and decreased female fertility compared to the wild-type zebrafish. These phenotypes can be rescued by increased food intake[63]. Perturbation of electron transfer efficiency by downregulating SCAF1 specifically affects hypoxic pancreatic cancer, breast and endometrial cancer proliferation, metabolism, and in vivo tumor growth[42,43]. Furthermore, immunohistochemical staining analysis of MDA in pancreatic tumor tissues exhibited an increase in MDA in SCAF1-downregulated xenografts versus vector-transduced xenografts[42]. However, there were no differences in Ki67 staining between the two xenografts[42]. These data indicate that impairment of supercomplex formation promotes ROS-mediated cell damage and cell death due to compromised ETC efficiency. We assume that SOX13/SCAF1 may modulate ferroptosis sensitivity by regulating mitochondrial biogenetics. NADPH ranked as the top dysregulated metabolite in SOX13-depleted RSL3$^{resis}$ SNU-668 cells, and NADPH abundance strongly correlates with sensitivity to ferroptosis inducers[44]. The hypothesis was validated experimentally.

Currently, research on ferroptosis-sensitivity related pharmacologically targetable biomarkers has just started[64–66]. Considering the important role in ferroptosis-related therapeutic efficacy, SOX13 might be an attractive anti-cancer drug target. We integrated molecular docking simulation with various in vivo and in vitro screening experiments, we selected zanamivir as the ferroptosis-enhancing agent. Our data showed that zanamivir can increase sensitivity to FINs of GC without toxicity. We revealed that zanamivir directly targets SOX13 to enhance its degradation and identified TRIM25 as a SOX13 interacting E3 ligase. Zanamivir was used at 10 mg by inhalation via the Diskhaler twice daily, or 10 mg inhaled plus 6.4 mg intranasally two or four times daily, for 5 days for treatment of influenza in clinical practice. In our paper, we found Zanamivir 5 mg/kg/d was sufficient to reverse the ferroptosis-resistant phenotype. In a number of studies, zanamivir was administrated intraperitoneally 10 mg/kg/d[67], 30 and 100 mg/kg/d[68], and 100 and 300 mg/kg/d, i.p[69]. in mice. And the used dosage was safe. More research should be conducted to explore how these dosages used in mouse relate to the likely maximum tolerated dosages that could be achieved in humans.

One limitation of this study is that our current knowledge of the modulation of the biogenesis of SCs and its biological functions is incomplete. This is possibly because of the limited availability of precise pharmacological and genetic tools to manipulate SCs formation.

Together, we report a pharmacologically targetable mechanism that drives resistance to FINs. We demonstrate that SOX13 directly targets SCAF1 and suppresses ferroptosis possibly via inducing the production of NADPH, which is a potential therapeutic strategy to increase chemosensitivity for advanced GC patients. However, the precise molecular mechanism requires further investigation.

# Methods

## Ethics declarations

Ethical clearance for all procedures involving human specimens was obtained from the local ethics committees of Yijishan Hospital, Wannan Medical College, and Fudan University Shanghai Cancer Center. Written informed consent was acquired from all participants providing human tumor tissues. Animal experiments strictly adhered to protocols approved by the Animal Care and Use Committee of Wannan Medical College.

## Reagents

Erastin (S7242), RSL3 (S8155), cis-diammineplatinum (II) dichloride (cisplatinum, S1166), IKE (S8877), doxorubicin (Adriamycin) HCl (S1208), gemcitabine (S1714), ferrostatin-1 (Fer-1, S7243), Z-VAD-FMK (S7023), and necrosulfonamide (NSA, S8251) were purchased from Selleck Chemicals (Houston, Texas, USA). The specific concentrations used in the study are detailed in the figure legends.

## Cell culture

Human GC cell lines were purchased from the American Type Culture Collection (ATCC). The mouse GC cell line (YTN16) was a gift from the laboratory of Jun Yu (The Chinese University of HongKong, HongKong)[70]. Short tandem repeat profiling and evaluation of isozymes, cell viability and mycoplasma were routinely performed[10,15,16]. We performed the last cell characterization in June 2023. The human GC cell lines SNU-484 and SNU-668 and the mouse GC cell line YTN16 were cultured with Roswell Park Memorial Institute 1640 medium (RPMI-1640, Corning, NY, USA) supplemented with 10% Fetal Bovine Serum (FBS, Gibco BRL, Grand Island, NY, USA) at 37 °C and 5% $CO_2$. GC cells were inoculated in a 10-cm cell culture dish and then subcultured into six-well plates for lipid peroxidation measurements and 96-well plates for cell viability measurements. With respect to the ferroptosis sensitivity test, cells were pretreated with DMSO or zanamivir (0 μM, 5 μM, 10 μM) for 12 h before exposure to the indicated concentrations of Erastin (0, 0.5, 1, 2, 5, and 10 μM) or RSL3 (0, 0.05, 0.1, 0.2, 0.5, 1, and 2 μM).

## Establishment of ferroptosis-resistant GC cells

Similar to previously reported procedures[12], SNU-484 or SNU-668 cells ($2 \times 10^6$) were inoculated in a 10-cm cell culture dish and exposed to low concentration of RSL3 (0.5 μM) or Erastin (2 μM) for 28 days, then these cells were exposed to high concentration of RSL3 (1 μM) or Erastin (5 μM) for another 28 days. The culture medium was routinely changed every other day, and the concentrations of the compounds were maintained at the indicated levels. The surviving clones were trypsinized, collected and expanded. A cell viability assay was utilized to assess the sensitivity of the surviving cells to Erastin or RSL3. The acquired ferroptosis-resistance was validated.

## Patients and samples

This study enrolled two independent cohorts consisting of 161 GC patients. In cohort 1, 109 fresh GC tissue pairs were collected from

patients who underwent gastrectomy in Shanghai Cancer Center (Fudan University) between 2008 and 2012[10,41]. The patients were treatment-naïve, including preoperative radio- or chemotherapy. Patients lost to follow-up were excluded. The patients had stage IIIA-C disease based on the AJCC-TNM staging system (8th edition). All patients received postoperative cisplatin-based adjuvant chemotherapy. In cohort 2, 52 endoscopic biopsy GC tissue pairs were obtained prior to cisplatin-based neoadjuvant chemotherapy in Yijishan Hospital (Wannan Medical College) between March 2019 and November 2020[10]. The tumor tissues were frozen in liquid nitrogen instantly after removal and stored at −80 °C. Patient clinical information is shown in Supplementary Tables 1 and 2. The collected data does not include any elements that could help to identify patients. The nontumorous samples were taken more than 3 cm away from the tumor. The tissues were confirmed by pathologists.

## Follow-up investigation

Regular follow-up assessments were conducted on GC patients at varying intervals post-surgery. Imaging studies, including chest X-ray and abdominal ultrasonography, as well as measurement of serum tumor markers (CA199, CEA, CA72-4, and CA125), were performed as part of the routine follow-up. Magnetic resonance imaging or contrast-enhanced CT scans of the upper abdomen were conducted every 3 months. Suspected cases of recurrence prompted immediate contrast-enhanced CT scans. Recurrence was comprehensively diagnosed based on radiological and histopathological findings. Disease-free survival (DFS) and overall survival (OS) were defined as the time interval between surgery and tumor recurrence diagnosis or last observation and the time from surgery to death or last follow-up, respectively. Follow-up data were retrieved from medical records or through direct communication with patients.

## Tissue microarray and immunohistochemistry

Immunohistochemical staining of tissue microarrays (TMAs) was carried out utilizing Envision TM + Dual Link System-HRP methods. To minimize nonspecific staining, slides were pre-incubated with appropriate preimmune serum. Following overnight incubation with primary antibodies against SOX13 (1:100)/SCAF1 (1:100) at 4 °C, slides were washed with phosphate-buffered saline (PBS) and then incubated with labeled polymer-HRP. Color reaction was initiated using 3,3′-diaminobenzidine tetrachloride (DAB) chromogen solution, followed by counterstaining with hematoxylin. Staining intensity and extent were independently assessed by two evaluators at 200 × magnification, and scores were assigned based on staining intensity (0-negative, 1-weak, 2-moderate, 3-strong) and extent (0%, 1–25%, 26–50%, 51–75%, 76–100%). The total score was a combination of intensity and extent scores. The co-expression of SOX13/SCAF1 was calculated by multiplying the immunohistochemical scores for both targets and dividing the samples into low and high co-expression groups based on the median value. Evaluation between evaluators showed a 94% agreement, and any scoring discrepancies were resolved through discussion.

## Tumor regression grade (TRG) and Becker criteria

The quantification of tumor regression grade (TRG) involved applying the Becker regression criteria. These criteria are centered on assessing the proportion of viable tumor cells in relation to the macroscopically identified tumor site, delineating four distinct categories: TRG1a (indicative of complete pathological regression with no residual tumor cells), TRG1b (representing subtotal regression with less than 10% residual tumor cells), TRG2 (illustrating partial regression with 10–50% residual tumor cells), and TRG3 (indicating minimal or no regression with over 50% residual tumor cells). Identification of the tumor bed was primarily based on unmistakable signs of tumor regression, such as notable fibrosis, necrosis, mucosal flattening, or presence of

macrophages. The extent of viable tumor cells was determined and correlated semiquantitatively to the tumor bed as a percentage value. Patients classified under TRG1a, TRG1b, or TRG2 categories were considered to be pathologically responsive.

## CRISPR−Cas9-mediated gene knockout

The application of CRISPR−Cas9 technology facilitated the targeted knockout of SCAF1. Guide RNAs were integrated into the pSpCas9 (BB)-Puro plasmid (Addgene, 62988) for this purpose. Following transfection of gastric cancer (GC) cells with the plasmid using Lipofectamine 3000, a subsequent selection process with 10 μg/ml puromycin (Santa Cruz Biotechnology) for 72 h was conducted. Single-cell clones showing successful knockout were chosen and expanded for further analysis. Guide RNA sequences to target human SCAF1 and mouse SCAF1 were meticulously designed and used for knockout clone identification through immunoblotting. Guide RNA sequences to target human SCAF1 were: GGTGTGGCAAATATGATAGG; Guide RNAs to target mouse SCAF1 were: GGTGACGATAGCCCCCGCC. Multiple deficient clones were pooled for subsequent experimental procedures.

## Plasmid construction

Standard procedures outlined in previous research studies were adhered to for plasmid construction. The primers are listed in Supplementary Table 3. For example, the SCAF1 promoter construct, (−1443/+60) SCAF1, was generated from human genomic DNA. This construct corresponds to the sequence from −1443 to +60 (relative to the transcriptional start site) of the 5′-flanking region of the human SCAF1 gene. It was generated with forward and reverse primers incorporating *KpnI* and *HindIII* sites at the 5′ and 3′-ends, respectively. The polymerase chain reaction (PCR) product was cloned into the *KpnI* and *HindIII* sites of the pGL3-Basic vector (Promega). The 5′-flanking deletion constructs of the SCAF1 promoter, (−926/+60) SCAF1, (−323/+60) SCAF1, and (−103/+60) SCAF1, were similarly generated using the (−1443/+60) *SCAF1* construct as a template. The SOX13 binding sites in the SCAF1 promoter were mutated using the QuikChange II Site-Directed Mutagenesis Kit (Stratagene). These constructs were meticulously confirmed via DNA sequencing to ensure accuracy and reliability.

Other promoter constructs were cloned in the same manner.

## Construction of lentivirus and stable cell lines

The generation of lentivirus and stable cell lines was conducted as per established protocols. Notably, for SOX13 overexpression, cDNA encoding SOX13 was PCR-amplified and subcloned into pGC-LV vectors. The resulting lentivirus containing the SOX13 gene was harvested, concentrated, and utilized to infect GC cells. Verification of SOX13 expression in infected cells was carried out through RT-PCR assessments. Additionally, stable knockdown of SOX13 was achieved in GC cell lines via lentiviral-based shRNA delivery using specific targets designed to suppress the gene expression, ensuring precise modulation of SOX13 levels.

**Transient transfection.** Cells were transfected using Lipofectamine 3000 with expression vector plasmids, promoter reporter plasmids, and control plasmids. Following transfection and recovery, serum-starved cells were prepared for subsequent assays, adhering to standardized transfection procedures.

## RNA preparation and qRT-PCR

Total RNA extraction from tissues and cells utilized Trizol reagent (Invitrogen, Carlsbad, CA, USA) following the manufacturer's protocol. The quality assessment of total RNA was conducted by determining the A260/A280 ratio through 1% agarose gel electrophoresis. Complementary DNA (cDNA) was synthesized using the GoScript Reverse Transcription System (Promega, Madison, Wis). Real-time RT-PCR was

performed with the SYBR Premix Dimmer Eraser kit (TaKaRa) using the ABI7500 system (Applied Biosystems, Carlsbad, CA, USA). β-Actin expression was used as the normalization control for gene expression analysis. Reactions were performed in triplicate, and the relative expression fold change of mRNAs was calculated utilizing the $2^{-\Delta\Delta Ct}$ method. Refer to Supplementary Table 3 for the primer details.

## Immunoblotting analysis

Cell lysates were prepared using RIPA cell lysis buffer (Cell Signaling Technology) supplemented with protease inhibitors after centrifugation. Protein content was determined using the Bradford assay, with 30–50 μg of proteins loaded onto an 8-12% SDS-polyacrylamide gel for electrophoresis and subsequently transferred to a PVDF membrane (Millipore). Following blocking with 5% skim milk in TBST, membranes were probed with primary antibodies overnight at 4 °C, washed, and then incubated with secondary antibodies. Detection of antibody-bound proteins was accomplished using ECL Western Blotting Substrate (Pierce, Rockford, IL), and band intensity analysis was performed with Image J software. See Supplementary Table 4 for antibody information.

## Chromatin immunoprecipitation

GC cells were serum-starved before cross-linking chromatin with 1% formaldehyde. Sonication was carried out to fragment chromatin, and immunoprecipitation was performed overnight at 4 °C with specific antibodies or isotype rabbit IgG. Following isolation of chromatin-antibody complexes using Protein A/G PLUS Agarose (Santa Cruz), DNA was purified after reversing crosslinks and analyzed using PCR to examine specific sequences from immunoprecipitated and input DNA. Results represent a minimum of three independent experiments.

## CHIP-Seq and data processing

Library preparation involved converting 10 ng DNA of each sample to phosphorylated blunt-end fragments, followed by Illumina adapter ligation and PCR enrichment. Sized products (-200–1500 bp) were selected with AMPure XP beads, denatured for single-stranded DNA generation, and subsequently sequenced on an Illumina NovaSeq 6000 platform (2 × 150 cycles). Sequencing data were processed using Off-Line Basecaller software for image analysis and base calling, followed by alignment to the human reference genome UCSC HG19 using BOWTIE (V2.1.0). MACS V1.4.2 was employed for peak calling of ChIP regions, identifying statistically significant enriched peaks with a $p$-value threshold of $10^{-4}$. Peaks were annotated based on the nearest gene using the UCSC RefSeq database, and data are accessible via GEO: GSE247870.

## Co-immunoprecipitation (co-IP)

Following a preliminary step involving IgG prewashing (Santa Cruz Biotechnology) and incubation with protein A/G Sepharose beads (Invitrogen) for 1 h at 4 °C, cell supernatants were subjected to overnight incubation with the appropriate primary antibody at 4 °C, followed by a 4-h incubation with protein A/G Sepharose beads. Subsequently, immunoprecipitated proteins were eluted and identified through Western blotting analysis.

## Cell viability and cytotoxicity

Cell viability was assessed using the CellTiter-Glo luminescent cell viability assay (Promega) as per the manufacturer's instructions. CellTiter-Glo substrate (15 μl) was introduced to cells cultured in a 96-well plate with 100 μl media, followed by 10 min of shaking and signal intensity quantification using a chemiluminescence plate reader. CellTox Green assay (Promega) involved the addition of dye (1:1000) to the media for cell death quantification through a fluorescence plate reader.

## Cell apoptosis assay

A total of $2 \times 10^5$ cells were stained with FITC Annexin V and PI from an apoptosis detection kit (BD, 556547). Subsequently, apoptosis analysis was carried out using a Beckman FC500 flow cytometer, with data analyzed using the FlowJo software package.

## Cell cycle analysis

Cell cycle analysis was executed using the BD Pharmingen BrdU FITC Flow kit (559619) in accordance with the manufacturer's guidelines.

## Lipid peroxidation C11-BODIPY assay

Cells were seeded in six-well plates at a density of $2 \times 10^5$ cells per well. Upon treatment with specific compounds, cells were collected, resuspended in 1 ml Hanks balanced salt solution (HBSS, Gibco) containing 10 μM BODIPY 581/591 C11 (Invitrogen, D3861), and incubated for 15 min at 37 °C. Following incubation, cells were washed and resuspended in 200 μl fresh HBSS, analyzed immediately with a flow cytometer for the assessment of lipid peroxidation. For BODIPY 581/591 C11 staining, the signals from both non-oxidized (phycoerythrin channel) and oxidized (FITC channel) C11 were monitored. The FITC to phycoerythrin mean fluorescence intensity ratio was calculated to identify cells undergoing lipid peroxidation. The gates to define what constituted an increased FITC/phycoerythrin fluorescence ratio were set based on untreated cancer cells, a condition that represents cells with little or no lipid peroxidation. At least 1000 cells were analyzed in each group and all experiments were repeated at least three times.

To quantify the lipid peroxidation in samples from animals that received immunotherapy, a single cell suspension was first prepared. Subcutaneous tumor tissue was resected and cut into small pieces, then tumor tissue was mechanically minced against a 100 μM cell strainer, and washed with PBS. The cell mixture was collected. Tumor and immune cells were pre-enriched using density gradient centrifugation (Ficoll, Sigma-Aldrich). The cell pellet was stained with anti-CD45 antibody, followed by BODIPY 581/591 C11. Cells were strained through a 40-μM cell strainer and analyzed immediately with a flow cytometer

## MDA measurement

The concentration of MDA in GC cells was determined using an Enzyme-linked immunosorbent assay (ELISA) kit (Abcam, ab118970) following the manufacturer's instructions.

## NADPH level and NADPH/NADP$^+$ ratio measurement

Intracellular NADPH levels and NADPH/NADP+ Ratio were measured in treated GC cells using the NADPH/NADP+ assay kit (S0179, Beyotime) according to the manufacturer's instructions.

## intracellular iron level, GPX4 specific activity and thiol measurement

The concentrations or activity of iron were measured using commercially available enzyme-linked immunosorbant assay (ELISA) kits (Abcam, ab83366). GPX4-specific activity was determined by measuring phosphatidylcholine hydroperoxide as substrate. To assay total mercaptans released into the medium, $2 \times 10^5$ cells per well were plated onto a six-well plate and cultured overnight. Subsequently, the medium was removed, and cells were washed two times with PBS and covered in serum-free and phenol-red-free medium. Aliquots were subsequently collected at desired time points. Total mercaptans secreted into the cell culture medium were determined using GSH as a standard.

## Metabolite extraction

For untargeted metabolomics, a total of 24 samples were analyzed ($n = 12$ Erastin$^{resis}$ SNU-668 cells transfected with shRNA-NC, $n = 12$ Erastin$^{resis}$ SNU-668 cells transfected with shRNA-SOX13). $2 \times 10^5$ cells

of adherent cells were harvested in six-well plates. When collected, cells were washed by cold PBS buffer twice and immediately quenched in liquid nitrogen. Tumor samples were weighed and pulverized. All samples were lysed in 1 ml of −80 °C extraction solvent (80% methanol/water). After centrifugation (20,000 g, 4 °C, 15 min), supernatant was transferred to a new tube, and samples were dried using a vacuum centrifugal concentrator. Blood samples from patients and mice were collected into BD Vacutainer blood collection tubes and placed on ice. Serum was isolated by centrifugation (15,000 g, 4 °C, 10 min), and aliquots of 100 μl of supernatant were frozen immediately at −80 °C. Metabolites were reconstituted in 150 μl of 80% acetonitrile/water, vortexed, and centrifuged to remove insoluble material. All samples were stored at −80 °C before LC-MS/MS analysis.

### Liquid chromatography-mass spectrometry and metabolomics data analysis

Liquid chromatography mass-spectrometry (LC-MS) and metabolomics data analysis was performed by Aksomics, biotech, shanghai, China.

For untargeted metabolomics, the ExionLC AD ultraperformance LC system (AB SCIEX) was coupled to a TripleTOF 5600 Plus mass spectrometer (AB SCIEX) for metabolite profiling. An ethylene bridged hybrid (BEH) amide column (100 × 2.1 mm, 1.7 μm, Waters) was used for metabolite separation. The quadrupole time-of-flight.

(QTOF)−MS is equipped with an electrospray ionization (ESI) probe with related parameters set as follows: source temperature, 550° (ESI+) and 450 °C (ESI−); ion spray voltage, 5500 (ESI+) and −4500 V (ESI−); atomization gas pressure, 55 psi; auxiliary heating gas pressure, 55 psi; curtain gas pressure, 35 psi. For targeted metabolomics,

the ExionLC AD ultraperformance LC system (AB SCIEX) was coupled to the Triple Quad 6500 triple-quadrupole MS/MS (AB SCIEX) for targeted analysis. A BEH hydrophilic interaction LC column (100 × 2.1 mm, 1.7 m, Waters) was used for the separation. The triple-quadrupole (QQQ)−MS/MS parameters were set as follows: source temperature, 550 °C; ion spray voltage, 5500 V; curtain gas, 35 psi; ion source gas 1 (nebulizer) and 2 (turbo ion spray), 55 psi.

Extraction and integration of LC-MS peak information were conducted using MarkerView 1.2.1.1 software (AB SCIEX). Following peak extraction, matching, alignment, and normalization preprocessing, the dataset matrix obtained was utilized for multidimensional statistical analysis using SIMCA-P 14.0 software (Umetrics). Data analysis encompassed multivariate and univariate statistical approaches. Metabolites were identified based on VIP (variable importance in the projection) > 1 and |pcorr| > 0.52 in multivariate models, with significance indicated by P < 0.05 from two-tailed Student's t test. Hierarchical clustering was executed through MetaboAnalyst 4.0 (Genome Canada), accessible online at http://www.metaboanalyst.ca.

### RNA-sequencing analysis

RNA-sequencing analysis was conducted by Aksomics, a biotech company in Shanghai, China, following established protocols. Total RNA samples were extracted using an RNA extraction kit (Takara) in adherence to the manufacturer's guidelines. Subsequently, libraries were sequenced on an Illumina HiSeq 4000 platform. Following quality control of raw reads, clean reads were aligned to the human genome using default parameters. Differentially expressed gene (DEG) normalized read counts, quantified as fragments per kilobase of exon per million (FPKM), were computed utilizing RSEM (v1.2.8). The DEG analysis and enrichment analysis were executed with Dr. TOM (BGI), an online software tool tailored for the differential expression analysis of RNA-seq data. The Kyoto Encyclopedia of Genes and Genomes (KEGG) pathways and Gene Ontology (GO) were annotated using the KEGG pathway database (http://www.genome.jp/kegg/) and the Gene Ontology Database (http://www.geneontology.org/), respectively. A false discovery rate (FDR)-corrected P value of 0.05 was deemed statistically significant.

### Oxygen consumption

Oxygen Consumption was determined as described previously[39]. For intact cells, $1.0 \times 10^4$ cells of the specified type were seeded in an XFE-24 Seahorse plate (Seahorse Biosciences, 102340) and left to adhere for 24 h at 37 °C with 5% $CO_2$. Media was then removed, and cells were washed with pre-warmed unbuffered RPMI-1640 without bicarbonate (Sigma, D5030) supplemented with 15 mM glucose, 2 mM sodium pyruvate and 1 mM glutamine. After the wash, 600 μL of the same buffer was added and cells were transferred to a 37 °C non-$CO_2$ incubator for 1 h. The Seahorse 24 optical fluorescent analyzer cartridge was prepared in the interim by adding 5 μM oligomycin, 0.5 μM FCCP, and 2 μM rotenone to each cartridge port. Oxygen consumption rates (pmol/min) were then measured using the Seahorse Bioanalyzer instrument at 37 °C. Subsequently, protein concentrations were determined using a BCA assay for normalization of OCR measurements.

In isolated mitochondria: To minimize variability between wells, mitochondria were first diluted 10× in cold 1x MAS (70 mM sucrose, 220 mM mannitol, 10 mM $KH_2PO_4$, 5 mM $MgCl_2$, 2 mM HEPES, 1.0 mM EGTA and 0.2% (w/v) fatty acid-free BSA, pH 7.2 at 37 °C). Stock substrates 0.5 M malic acid, 0.5 M pyruvic acid or 0.5 M succinate and 0.2 mM ADP, were subsequently diluted to the concentration required for plating. Next, while the plate was on ice, 50 μL of mitochondrial suspension (containing 25 μg of mitochondria) was delivered to each well (except for background correction wells). The XF24 cell culture microplate was then transferred to a centrifuge equipped with a swinging bucket microplate adaptor, and spun at 2000 g for 20 min at 4 °C. The Seahorse 24 optical fluorescent analyzer cartridge was prepared in the interim by adding 4 mM ADP and 10 μM rotenone to each cartridge port.

After centrifugation, 450 μL of prewarmed (37 °C) 1X MAS+ substrates Pyruvate/malate (10 mM/2 mM) or succinate (10 mM) were added to each well. In the case of succinate driven respiration, 100 μM Rotenone was also added to the MAS buffer. The mitochondria were viewed briefly under a microscope at 20× to ensure consistent adherence to the well. The plate was then transferred to the Seahorse XFe/XF24 Analyzer, and the experiment initiated.

### Complex activities

Complex activities were determined as described previously[39]. CIV activity was measured spectrophotometrically with 10 μg of isolated mitochondria. 1 mg/ml of reduced cytochrome C (Sigma, C2506) was added and the decrease in absorbance at 550 nm was measured for 3 min at 37 °C in a 96-well plate. Rotenone sensitive NADH-dehydrogenase activity (CI activity) was measured at 340 nm ($\varepsilon = 6.22 \text{ mM}^{-1}\text{cm}^{-1}$) in a mix containing buffer CI/CII (25 mM $K_2HPO_4$ pH 7.2, 5 mM $MgCl_2$, 3 mM KCN, 2.5 mg/ml BSA), 0.13 mM NADH, 0.13 mM UQ1, and 0.2 μg/ml antimycin A. Rotenone sensitivity was measured under the same conditions with the addition of 5 μM rotenone. For complex II measurements, succinate dehydrogenase activity (CII activity) was measured using 100 mg of isolated mitochondrial resuspended in 950 mL CI/CII buffer (25 mM $K_2HPO_4$ pH 7.2, 5 mM $MgCl_2$, 3 mM KCN, 2.5 mg/ml BSA, 10 mM succinate, 0.03 mM DCPIP, 2 μg/ml antimycin A and 5 mM rotenone). Isolated mitochondria were incubated for 10 min at 37 °C. The reaction was initiated by adding 15 μL of 10 mM ubiquinone, and the decrease in absorbance at 600 nm was measured for 4 min at 37 °C in a 96 well plate spectrophotometrically. CI + III activity was measured at 550 nM incubating mitochondria in buffer containing 25 mM $K_2HPO_4$ pH 7.2, 5 mM $MgCl_2$, 0.5 mM, 2.5 mg/ml BSA, 1 mg/ml oxydized cytochrome C and 1 mg/ml NADH. For CII + III a buffer containing 25 mM $K_2HPO_4$ pH 7.2, 5 mM $MgCl_2$, 0.5 mM KCN, 5 μM Rotenone 2.5 mg/ml BSA, 1 mg/ml oxidized cytochrome C and 1 mg/ml Succinate, was used. Lastly, Citrate synthase (CS) activity was measured resuspending 10 mg of isolated mitochondrial in buffer containing 10 Tris (100 mM, pH 8.0), 10 μM of DTNB and Ac-CoA (30 μM). Reaction was started by adding 50 mL of

10 mM oxaloacetic acid and the increase in absorbance at 412 nm was monitored for 3 min.

## Molecular docking modeling assay

Molecular docking modeling assay was performed as described previously[71]. The 3D structure SOX13 were downloaded from the AlphaFold (https://alphafold.ebi.ac.uk/entry/Q9UN79). The 3D structure of sequence regions with active protein structure (HMG Box, 424aa-494aa) of the SOX13 protein was optimized using Molecular dynamics (MD) simulation software (Desmond v3.8 modules in Schrödinger suit). In the MD simulation, an appropriate amount of $Na^+$ counter-ions were used to neutralize the charges in the complexes. To evaluate the stability of the SOX13 protein, the protein was first minimized and subjected to MD in the NVT ensemble. The temperature of the system was raised from 0 to 300 K. After the initial equilibration, a molecular dynamics production run was carried out for 200 ns. After 200 ns of MD simulation, the established 3D structure was finally evaluated by the Procheck and Ramachandran Plot analysis. The screening compound database contains 2000,000 compounds. For the database to be screened, these structures were prepared by the LigPrep module in the Schrodinger software package to generate 3D conformations and set in the protonation state of pH 7, while its conformation was optimized under the OPLS-3e force field. In structural molecular biology and computer-assisted drug design, molecular docking is a key computer-based technique, which is used to investigate the predominant binding mode between a ligand and a protein of known 3D structure or, in other words, a receptor. To explore the interaction mechanism and potential binding sites of SOX13 protein with compounds, the molecular docking between SOX13 and the compounds was determined by using glide software, which is a module of Maestro/Schrödinger's molecular modeling software package. The active sites of the 3D structure of the SOX13 protein were predicted using SiteID software. The parameters of the active site were set as follows: Center_x = −2.163, center_y = 8.820, and center_z = 4.8253. Finally, conformations were obtained and ranked by glide score, which is an empirical scoring function that combines multiple parameters. The unit of the glide score was kcal/mol. The optimal active site was selected based on the glide score for further analysis, in order to find potential drug targets.

## Cellular thermal shift assay

To determine target engagement of SOX13 by zanamivir within cells, Erastin[resis] SNU-668 cells with 70–80% confluence in 15 cm culture dish were treated with zanamivir or vehicle (DMSO) for 1 h. Cells were harvested and washed once with PBS, then suspended in 1 ml of PBS supplemented with proteinase and phosphatase inhibitors (Beyotime, Shanghai, China) and also maintained with the same dose of zanamivir or DMSO as initial treatment. The cell suspension was distributed into seven 0.2 ml PCR tubes at different designated temperature. Samples were heated at different designated temperature for 2 min using a 96-well thermal cycler. Tubes were removed and incubated at room temperature for 3 min immediately after heating. Three freeze and thaw cycles in liquid nitrogen were performed to lyse the cells. The tubes were vortexed briefly after each thawing. The cell lysate was collected and cell debris together with precipitated and aggregated proteins was removed by centrifuging samples at 20,000 $g$ for 20 min at 4 °C. Cell lysate samples were boiled for 5 min at 95 °C with loading buffer and subjected to western blotting analysis.

## Surface plasmon resonance analysis

The investigation into the binding affinity between zanamivir and SOX13 protein involved the utilization of Biacore 8 K along with the Biacore Insight Evaluation software. Initially, purified SOX13 protein (0.17 mg/mL) was dissolved in PBS and then immobilized onto the CM5 chip (GE Healthcare, USA). Various concentrations of zanamivir, dissolved in running buffer (1 × PBS with filtration, 2% DMSO), were subsequently passed over the chip to generate response signals. The kinetics and affinities were assessed using the Biacore Insight Evolution software, ultimately yielding the binding affinity value (Kd).

## In vivo xenograft assays

Female BALB/c nu/nu athymic and C57BL/6 mice were purchased from the Experimental Animal Center of the Chinese Academy of Sciences (Shanghai, China) and were 5–6 weeks old. The nude mice were held under specific-pathogen-free housing conditions with food and water being supplied ad libitum. The temperature was maintained at 23 °C ± 2 °C, humidity 40–60%, and 12 h light/dark cycles. In CDX studies, SNU-668 cells ($2 \times 10^6$ cells suspended in 100 μl PBS) were injected subcutaneously into nude mice. On Day 7 after injection, the tumor grew to ~80 mm³, and the mice were divided into different treatment groups randomly ($n = 4$ mice per group) and then treated with cisplatin (4 mg/kg, i.p., once weekly), IKE (20 mg/kg, i.p., once daily), zanamivir (5 mg/kg, i.p., once daily) or PBS (100 μl, i.p., once daily) for 3 consecutive weeks starting on Day 7. Mice were euthanized by $CO_2$ asphyxia when the experiments ended or the longest dimension of the tumors reached 2.0 cm. The tumors were excised, weighed, and then stored at −80 °C until further analysis.

To clarify the influence of SOX13/SCAF1 on the efficacy of immunotherapy, $2 \times 10^6$ YTN16 cells transfected with the desired vector were subcutaneously inoculated into C57BL/6 mice. On Day 7 after injection, mice were divided into two treatment groups randomly and afterwards were treated with isotype (100 μg, i.p., once every 3 days) or PD1 (BioXCell, BE0273, 29 F.1A12, 100 μg, i.p., once every 3 days) for 21 consecutive days. Excised tumors were enzymatically digested and processed according to manufacturer protocol utilizing the Tumor Dissociation Kit (130-096-730, Miltenyi Biotec, Germany) at 37 °C for 45 min prior to immune profiling by FACS. The maximal tumor size/burden permitted by our ethics committee review board was 2000 mm³. We confirm that none of the mice included in this study exceeded this limit. Tumor volume was calculated with the formula length × width²/2 using a digital caliper.

## In vivo toxicity

Four-week-old female BALB/c mice were randomly assigned to three groups ($n = 4$) for toxicity assessment. The groups were administered DMSO (i.p., Group 1), or zanamivir at 5 mg/kg/day (i.p., Group 2), and 10 mg/kg/day (i.p., Group 3) doses. Daily records of each mouse's body weight were maintained throughout the 28-day experimental period. Following this, mice were euthanized, and selected tissues were fixed in 4% paraformaldehyde. Their hearts, lungs, livers, spleens, and kidneys were sectioned and stained with H&E for histological analysis.

## Statistical analysis

Data are presented as the mean ± standard deviation of three or four biologically independent experiments. The data were analyzed with GraphPad Prism 8.0 (GraphPad Software, La Jolla, CA). Error bars are not visible in cases where the error is relatively smaller than the symbol. Statistical differences between pairs of groups were analyzed by independent two-tailed Student's $t$-test. Survival differences were assessed by Kaplan–Meier methods, and statistical significance was determined by the log-rank test. The correlation between SOX13/SCAF1 coexpression scores and tumor regression grade was estimated with the nonparametric Mann-Whitney Wilcoxon test. The correlation between SOX13 and SCAF1 expression levels was determined using Pearson correlation analysis. Statistical significance was set to $p < 0.05$ and represented as * $p < 0.05$, ** $p < 0.01$, *** $p < 0.001$, **** $p < 0.0001$, with ns as no significance, as indicated in the figure legends。

## Reporting summary

Further information on research design is available in the Nature Portfolio Reporting Summary linked to this article.

## Data availability

The Cancer Therapeutics Response Portal (portals.broadinstitute.org/ctrp/) compound sensitivity dataset, a data matrix containing the normalized AUC values of each compound in each cell line, was downloaded from [https://ocg.cancer.gov/programs/ctd2/data-portal][17–19]. Genome binding/occupancy profiling of SOX13 by high throughput sequencing has been deposited in Gene Expression Omnibus data base (GEO) under accession code GSE247870. The data of RNA-seq of ferroptosis-resistant GC cells generated in this study have been deposited in GEO under accession code GSE262114. The data of RNA-seq of GC cells transfected with shRNA-SOX13 or shRNA-NC have been deposited in GEO under accession code GSE211072. The data of untargeted metabolites of GC cells transfected with shRNA-SOX13 or shRNA-NC have been deposited in Metabolomics Workbench under accession code ST003134 [https://doi.org/10.21228/M82431]. The remaining data are available within the Article, Supplementary Information or Source Data file. Source data are provided with this paper.

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

## Acknowledgements

We thank the Department of Central Laboratory from Yijishan Hospital for technical assistance. We thank Dan Ye (Institutes of Biomedical Sciences, Shanghai Medical College, Fudan University) for providing suggestions and assistance during the preparation of this manuscript. We thank Dr. Mingjie Chen from NewCore Biotech Co., Ltd. Shanghai, for bioinformatics data analysis support.

## Author contributions

Conceptualization: Mingzhe Ma, Hui Yang, Kun Lv, Yan Zhang, and Yangbai Sun. Investigation, formal analysis, and methodology: Hui Yang, Qingqing Li, Xingxing Chen, Mingzhe Weng and Yakai Huang. Validation: Xiaocen Liu, Haoyu Huang, Mengying Zhang, Weiya Pei, Xueqin Li, Qingsheng Fu, and Liangyu Zhu. Review and editing: Mingzhe Ma, Hui Yang and Yanhuizhi Feng. Funding acquisition: Hui Yang, Xiaocen Liu, Kun Lv, Yan Zhang, and Mingzhe Ma. All authors approved the final version of the manuscript.

## Competing interests

The authors declare no competing interests. This work was supported by the following grants: M.Z.M. discloses support for the research of this work from National Nature Science Foundation of China [grant number 81972213] and Natural Science Foundation of Shanghai [grant number 22ZR1412700]. K.L. discloses support for publication of this work from the National Nature Science Foundation of China [grant number 82072370], Program for Excellent Sci-tech Innovation Teams of Universities in Anhui Province [grant number 2022AH010074] Funding of "Peak" Training Program for Scientific Research of Yijishan Hospital, Wannan Medical College [grant number GF2019T01] and Funding of Climbing Peak Training Program for Innovative Technology team of Yijishan Hospital, Wannan Medical College [grant number PF201904]. H.Y. discloses support for the research of this work from the Health Research Program of Anhui [grant number AHWJ2023A30252], National Nature Science Foundation of China [grant number 81802503], Open Project of Key Laboratory of Non-coding RNA Transformation Research of Anhui Higher Education Institution (Wannan Medical College) [grant number RNA202205], Science and Technology Application Basic Research Project of Wuhu [grant number 2022jc60], Funding of "Peak" Training

Program for Scientific Research of Yijishan Hospital, Wannan Medical College [grant number GF2019G15], Outstanding Innovative Research Team for Molecular Enzymology and Detection in Anhui Provincial Universities (2022AH010012). Y.Z. discloses support for the research of this work from the Key University Science Research Project of Anhui Province [grant number KJ2020A0594] and Key Health Research Project in Anhui Province [grant number AHWJ2022A021]. X.C.L discloses support for the research of this work from the Key University Science Research Project of Anhui Province [grant number KJ2020A0607]. Y.B.S. discloses support for the research of this work from Shanghai Municipal Health Commission Research Project [grant number 20194Y0242].

## Additional information

[1]Central Laboratory, The First Affiliated Hospital of Wannan Medical College (Yijishan Hospital of Wannan Medical College), Wuhu, Anhui, China. [2]Anhui Province Key Laboratory of Non-coding RNA Basic and Clinical Transformation, Wuhu, Anhui, China. [3]Anhui Provincial Key Laboratory of Molecular Enzymology and Mechanism of Major Diseases, College of Life Sciences, Anhui Normal University, Wuhu, Anhui, China. [4]Research Center of Health Big Data Mining and Applications, School of Medical Information, Wannan Medical College, Wuhu, Anhui, China. [5]Department of Radiation Oncology, Fudan University Shanghai Cancer Center, Shanghai, China. [6]Department of General Surgery, Xinhua Hospital, School of Medicine, Shanghai Jiaotong University, Shanghai, China. [7]Department of Gastric Surgery, Fudan University Shanghai Cancer Center, Shanghai, China. [8]Minimally Invasive Therapy Center, Department of Integrative Oncology, Fudan University Shanghai Cancer Center, Shanghai, China. [9]Department of Nuclear Medicine, The First Affiliated Hospital of Wannan Medical College (Yijishan Hospital of Wannan Medical College), Wuhu, Anhui, China. [10]Key Laboratory of Non-coding RNA Transformation Research of Anhui Higher Education Institution, Wannan Medical College, Wuhu, Anhui, China. [11]Clinical Research Center for Critical Respiratory Medicine of Anhui Province, Wuhu, Anhui, China. [12]Department of Implantology, Stomatological Hospital and Dental School of Tongji University, Shanghai Engineering Research Center of Tooth Restoration and Regeneration, Shanghai 200072, China. [13]Department of Gastrointestinal Surgery, The First Affiliated Hospital of Wannan Medical College (Yijishan Hospital of Wannan Medical College), Wuhu, Anhui, China. [14]Department of Gastroenterology, The First Affiliated Hospital of Wannan Medical College (Yijishan Hospital of Wannan Medical College), Wuhu, Anhui, China. [15]Department of Musculoskeletal Surgery, Fudan University Shanghai Cancer Center, Shanghai, China. [16]These authors contributed equally: Hui Yang, Qingqing Li, Xingxing Chen, Mingzhe Weng, Yakai Huang. ✉e-mail: lvkun315@126.com; yanyan0921@sina.com; drsunyb@fudan.edu.cn; mmz666@163.com

