## [Peer Review File · Nature Communications]

Reviewers' Comments:

Reviewer #1:

Remarks to the Author:

The revised manuscript has been significantly improved. Overall, the data from Fig. 1-4 and 6-8 are extensive and solid. However, significant concerns remain for fig. 5 (which studies the mechanisms by which SCAF1 regulates ferroptosis).

The authors propose that SCAF1 facilitates ferroptosis resistance by promoting the formation of the ETC complexes, particularly the interaction between complex III and IV, while concurrently elevating NADPH levels. While this reviewer acknowledges the intricate and context-dependent nature of ETC complex involvement in ferroptosis, this study falls short in elucidating a precise mechanism by which augmenting complex III-IV interaction via SCAF1 fosters resistance to ferroptosis.

Do they imply that the enhancement of complex III-IV interaction by SCAF1 suppresses ferroptosis by increasing NADPH generation (a seemingly improbable scenario)? Are the effects on complex III-IV formation and NADPH generation considered independent/parallel downstream consequences of SCAF1 (if so, the effect on NADPH production can already explain the ferroptosis suppressing function of SCAF1, and SCAF1's role in regulating complex formation might have little to do with its function in modulating ferroptosis), or does SCAF1 modulate NADPH production by regulating complex III-IV interaction (if so, how complex III-IV interaction affects NADPH production)?

Overall, the mechanisms underlying SCAF1's regulation of ferroptosis lack clarity and coherence based on the current data. Furthermore, the authors should add a schematic diagram to summarize their proposed model, which will help reviewers and readers to better comprehend their suggested mechanism.

They need to show western blotting data of SOX13 knockdown in cisplatin resistant cells (corresponding to Fig. S6E).

Reviewer #2:

Remarks to the Author:

The authors have adequately addressed my comments from the initial round of review.

Reviewer #3:

Remarks to the Author:

Major Comments

I had reviewed this paper previously. My comments relate to the current version

1) There are many important key points that are claimed but not substantiated. For example, (line 108) the authors write "GC is the second most insensitive tumor type to Erastin (Supplementary Figure 1A), suggesting that GC is relatively resistant to ferroptosis induction." How was this analysis performed, on what basis was this claim made, what are the numbers etc? Perhaps the data is presented in the Supplementary Information, but please do not expect a reader or a reviewer to flip back and forth between the different sections.

2) Likewise, there are many key facts that are not explained, making it very difficult for a reviewer to assess the reliability of the results. Specifically, what is Erastin? What is RSL-3? The first time these compounds are mentioned on line 112 and it is unlikely a reader not intimately knowledgeable about ferroptosis will be familiar with these compounds (unlike say well-known genes like P53). Why were two different compounds used? Do the cell lines show cross-resistance?

3) The association between high SOX13 expression and ferroptosis resistance should be validated in more cell lines that show endogenous sensitivity or resistance to ferroptosis.

4) The specificity of the SOX13-associated resistance could be further tested. It seems that high SOX13 imparts resistance to ferroptosis and cisplatin, but not apoptosis, doxorubicin, or gemcitabine. It would be useful to test a few more commonly used gastric cancer chemotherapies such as 5-FU, oxaliplatin, and taxanes

5) (line 203) "When we compared the commonly dysregulated genes detected by RNA-seq with SOX13 downregulation, the genes with SOX13-bound transcription sites characterized by ChIP-seq and the genes dysregulated in FIN-resistant cells, we identified one overlapping gene SCAF1 (Figure 3A)." This seems a bit strange to me in that the overlap of the three data sets only pinpoints one gene, and could this be random chance? This could be assessed in a permutation analysis.

6) (line 225) "According to JASPAR, we identified three potential SOX13 binding motifs in the promoter area of SCAF1 (2000 bp upstream of the 226 transcription start site) (Figure 3G)." Do these three potential SOX13 binding motifs show SOX13 Chip-seq signals?

7) Is the extent of resistance conferred by SOX13 and SCAF1 exactly the same? It seems a bit surprising that among the many hundreds of genes SOX13 must regulate (as a transcription factor) that **all** of the effect goes through a single gene (SCAF1). It would be useful to compare the resistance/sensitivity of SCAF1 knockdown vs SANF1 + SOX13 knockdown.

8) (line 316) "The increased NADPH production when acquiring ferroptosis-resistance was validated in vivo." Does this refer to Figure 5I? I do not think this is strong rigorous evidence, as changes in blood could be due to a wide variety of causes.

9) Overall, I find that this study suffers from too many claims, each supported by a fairly thin line of evidence. The study would be better off being broken up into two manuscripts – perhaps as a suggestion one focused on the role of SOX13/SCAF1 in ferroptosis, and another on the therapeutic mechanisms (immune blockade, zanamivir and its mechanism of action).

Minor Comments

1) I would suggest the manuscript should undergo some copyediting – phrases like "Ferroptosis has been linked to the efficacy of a lot of anticancer therapies," (line 96) would be better phrased as "multiple anticancer therapies". This will improve readability.

2) (Line 275) "To further consolidate the clinical relevance of ferroptosis to cisplatin-based chemotherapy in GC patients, we performed 4-HNE staining in pre- and post-chemotherapy GC samples in cohort 2." What is 4-HNE?

Fudan University Shanghai Cancer Center
复旦大学附属肿瘤医院

Ming-Zhe Ma, M.D., Ph.D.

A.P., Department of Gastric Surgery

Fudan University Shanghai Cancer Center and Cancer Institute

270 Dong-An Road, Shanghai 200032, P.R. China

Office: 86-21-64175590

Fax: 86-21-64434556

Email: mmz666@163.com

January 16th, 2024

Reviewer #1:

Major issue 1: The revised manuscript has been significantly improved. Overall, the data from Fig. 1-4 and 6-8 are extensive and solid. However, significant concerns remain for fig. 5 (which studies the mechanisms by which SCAF1 regulates ferroptosis).

The authors propose that SCAF1 facilitates ferroptosis resistance by promoting the formation of the ETC complexes, particularly the interaction between complex III and IV, while concurrently elevating NADPH levels. While this reviewer acknowledges the intricate and context-dependent nature of ETC complex involvement in ferroptosis, this study falls short in elucidating a precise mechanism by which augmenting complex III-IV interaction via SCAF1 fosters resistance to ferroptosis.

Do they imply that the enhancement of complex III-IV interaction by SCAF1

suppresses ferroptosis by increasing NADPH generation (a seemingly improbable scenario)? Are the effects on complex III-IV formation and NADPH generation considered independent/parallel downstream consequences of SCAF1 (if so, the effect on NADPH production can already explain the ferroptosis suppressing function of SCAF1, and SCAF1's role in regulating complex formation might have little to do with its function in modulating ferroptosis), or does SCAF1 modulate NADPH production by regulating complex III-IV interaction (if so, how complex III-IV interaction affects NADPH production)?

Overall, the mechanisms underlying SCAF1's regulation of ferroptosis lack clarity and coherence based on the current data.

Response: We thank the reviewer's helpful suggestion. SCAF1, supercomplex assembly factor 1, has been reported to promote structural attachment between III₂ and IV and thus increase NADH-dependent respiration.^[1] SCAF1-deficient cells exhibit decreased mitochondrial efficiency and NADH levels.^[1,2,3] The effect of SOX13/SCAF1 on the production of NADH was confirmed in Supplemental Figure 16. Consistent with the central role of NADPH in the cellular metabolic network and redox homeostasis, it has been recently discovered that NADPH could regulate ferroptosis from multiple aspects. For instance, Wang *et al.* discovered that oxidoreductases on the endoplasmic reticulum such as NADPH-cytochrome P450 reductase (POR) and NADH-cytochrome b5 reductase (CYB5R1) were responsible for the catalyzation of

lipid peroxidation during ferroptosis.^[4] Under normal conditions, POR and CYB5R1 would transfer the electrons of NADPH/NADH to the downstream proteins such as cytochromes P450.^[5] However, they may incidentally donate the electrons to the oxygen in the cellular environment under certain circumstances to generate H₂O₂, which would be further converted to hydroxyl radicals through the iron-catalyzed Fenton reaction route. Due to the high reactivity of the hydroxyl radicals, they could cause the abstraction of hydrogen atoms from the methylene carbons in the polyunsaturated fatty acids (PUFAs) and readily initiate lipid peroxidation, leading to the disruption of the plasmamembrane.^[6-8] NADPH could also act as electron carriers and donate electrons to enable the glutathione reductase-mediated reduction from glutathione disulfide (GSSG) to glutathione (GSH).^[9] Meanwhile, NADPH could also support the SLC7A11-mediated intake of cystine and further convert it into cysteine for GSH synthesis.^[10-13] GSH would subsequently participate in the glutathione peroxidase 4 (GPX4)-mediated elimination of lipid ROS to inhibit ferroptosis.^[12,14] In addition to GSH regeneration and synthesis, NADPH also participates in the thioredoxin reductase (TR)-mediated regeneration of thioredoxin (Trx) and form a functional system for the reduction of protein-disulfides.^[12,13] Specifically, the reduced Trx could be oxidized by donating a proton from the sulfhydryl group to reduce other protein-disulfides, and the oxidized Trx then reacts with NADPH under the catalyzation of TR and is converted back to the reduced form for recycling.^[12,13,15] This system is a critical biochemical component in redox homeostasis, antiviral defense, and ferroptosis regulation. For

instance, the report by Evijola Llabani *et al.* showed that a small-molecule compound ferroptocide could inhibit Trx in a targeted manner and initiate ferroptotic cell death.^[16] There are also reports that NADPH is involved in a non-GPX4-dependent ferroptosis route by cooperating with ferroptosis suppressor protein 1 (FSP1) to reduce CoenzymeQ10 (CoQ10) to CoQ10-H2, which is a potent antioxidant capable of preventing the propagation of lipid peroxidation on the plasma membrane.^[17,18] Mariluz Soula *et al.* demonstrated that NADPH could also affect the ferroptosis process by mediating the regeneration of BH4, which is an effective free radical trapping antioxidant capable of effectively clearing lipid peroxide to protect cells from ferroptotic death when GPX4 activity is inhibited. The underlying molecular mechanism is that NADPH could act as the cofactor to enable the regeneration of tetrahydrobiopterin (BH4) via difolate reductase (DHFR) in addition to the de novo guanosine triphosphate cyclohydrolase I (GCH1)-mediated synthesis route.^[19] Extending from the evidence above, it was discovered that inhibiting the NAD⁺ kinase (NADK) could reduce the cellular NADPH levels and enhance their susceptibility to ferroptosis induction by typical ferroptosis inducers such as erastin, RSL3, and FIN56, further validating the potential application of NADPH as a biomarker for assessing ferroptosis sensitivity.^[20,21] Recently, Leu *et al.* demonstrated that by replacing the proline (P47) at the codon 47 of the human p53 protein into serine (S47), the cells would show reduced glycolysis capacity while metabolism through PPP would be enhanced significantly, accompanied with increasing NADPH/NADP⁺ ratio.^[22] S47 cells showed

distinctive metabolic profile compared to P47 cells, characterized by lower ROS stress and expression of Activating Transcription Factor 4 (ATF4).^[22] However, enhancing the expression of ATF4 in S47 cells could reverse this trend and inhibit the PPP activities therein, leading to increasing sensitivity to ferroptosis-inducing treatments.^[22] Overall, it could be concluded that the NADPH could boost the antioxidant defense in tumor cells by supporting the biosynthesis of GSH, Trx, and CoQ10-H2 to rescue them from ferroptosis.

SCAF1 has been reported to modulate the level of NADPH.^[23] As NADPH ranked among the top three dysregulated metabolites in SOX13-depleted RSL3^{resis} SNU-668 cells and NADP(H) can be produced by phosphorylating NAD(H), we speculated whether SOX13/SCAF1 could regulate NADPH production through modulating NADH levels. We found that SOX13/SCAF1 silencing decreased the production of NADH as demonstrated in Supplemental Figure 16. Furthermore, ectopic expression of SOX13 in sensitive cells upregulated the levels of NADPH, and this effect was abolished by SCAF1 knockout (Supplemental Figure 16F). To verify the role of SOX13/SCAF1 in production of NADPH was dependent on its effect on NADH production, we knocked down NAD kinases. NAD kinases (NADKs) are the only enzymes that generate NADP(H) by phosphorylating NAD(H).^[24,25] We found that the cellular NADPH level induced by SOX13/SCAF1 was significantly decreased with silencing of NADK (Supplemental Figure 16H). In parallel with this notion, the survival benefits of SOX13/SCAF1-overexpressing GC cells were largely eliminated

with NADK silencing (Supplemental Figure 16I).

Reference

1. Calvo E, Cogliati S, Hernansanz-Agustín P, Loureiro-López M, Guarás A, Casuso RA, García-Marqués F, Acín-Pérez R, Martí-Mateos Y, Silla-Castro JC, Carro-Alvarellos M, Huertas JR, Vázquez J, Enríquez JA. Functional role of respiratory supercomplexes in mice: SCAF1 relevance and segmentation of the Qpool. *Sci Adv.* 2020;6(26):eaba7509.
2. Hollinshead KER, Parker SJ, Eapen VV, Encarnacion-Rosado J, Sohn A, Oncu T, Cammer M, Mancias JD, Kimmelman AC. Respiratory Supercomplexes Promote Mitochondrial Efficiency and Growth in Severely Hypoxic Pancreatic Cancer. *Cell Rep.* 2020;33(1):108231.
3. Ikeda K, Horie-Inoue K, Suzuki T, Hobo R, Nakasato N, Takeda S, Inoue S. Mitochondrial supercomplex assembly promotes breast and endometrial tumorigenesis by metabolic alterations and enhanced hypoxia tolerance. *Nat Commun.* 2019;10(1):4108.
4. Yan B, Ai Y, Sun Q, Ma Y, Cao Y, Wang J, Zhang Z, Wang X. Membrane Damage during Ferroptosis Is Caused by Oxidation of Phospholipids Catalyzed by the Oxidoreductases POR and CYB5R1. *Mol Cell.* 2021;81(2):355-369.e10.
5. Zou Y, Li H, Graham ET, Deik AA, Eaton JK, Wang W, Sandoval-Gomez G, Clish CB, Doench JG, Schreiber SL. Cytochrome P450 oxidoreductase contributes to phospholipid peroxidation in ferroptosis. *Nat Chem Biol.* 2020;16(3):302-309.

6. Doll S, Proneth B, Tyurina YY, Panzilius E, Kobayashi S, Ingold I, Irmeler M, Beckers J, Aichler M, Walch A, Prokisch H, Trümbach D, Mao G, Qu F, Bayir H, Füllekrug J, Scheel CH, Wurst W, Schick JA, Kagan VE, Angeli JP, Conrad M. ACSL4 dictates ferroptosis sensitivity by shaping cellular lipid composition. *Nat Chem Biol.* 2017;13(1):91-98.
7. Perez MA, Magtanong L, Dixon SJ, Watts JL. Dietary Lipids Induce Ferroptosis in *Caenorhabditiselegans* and Human Cancer Cells. *Dev Cell.* 2020;54(4):447-454.e4.
8. Lee JY, Nam M, Son HY, Hyun K, Jang SY, Kim JW, Kim MW, Jung Y, Jang E, Yoon SJ, Kim J, Kim J, Seo J, Min JK, Oh KJ, Han BS, Kim WK, Bae KH, Song J, Kim J, Huh YM, Hwang GS, Lee EW, Lee SC. Polyunsaturated fatty acid biosynthesis pathway determines ferroptosis sensitivity in gastric cancer. *Proc Natl Acad Sci U S A.* 2020;117(51):32433-32442.
9. Ballatori N, Krance SM, Notenboom S, Shi S, Tieu K, Hammond CL. Glutathione dysregulation and the etiology and progression of human diseases. *Biol Chem.* 2009;390(3):191-214.
10. Forman HJ, Zhang H, Rinna A. Glutathione: overview of its protective roles, measurement, and biosynthesis. *Mol Aspects Med.* 2009;30(1-2):1-12.
11. Liu X, Olszewski K, Zhang Y, Lim EW, Shi J, Zhang X, Zhang J, Lee H, Koppula P, Lei G, Zhuang L, You MJ, Fang B, Li W, Metallo CM, Poyurovsky MV, Gan B. Cystine transporter regulation of pentose phosphate pathway dependency and disulfide stress exposes a targetable metabolic vulnerability in cancer. *Nat Cell Biol.*

2020;22(4):476-486.

12. Liu X, Zhang Y, Zhuang L, Olszewski K, Gan B. NADPH debt drives redox bankruptcy: SLC7A11/xCT-mediated cystine uptake as a double-edged sword in cellular redox regulation. *Genes Dis.* 2020;8(6):731-745.

13. Seco-Cervera M, González-Cabo P, Pallardó FV, Romá-Mateo C, García-Giménez JL. Thioredoxin and Glutaredoxin Systems as Potential Targets for the Development of New Treatments in Friedreich's Ataxia. *Antioxidants (Basel).* 2020;9(12):1257.

14. Yang WS, SriRamaratnam R, Welsch ME, Shimada K, Skouta R, Viswanathan VS, Cheah JH, Clemons PA, Shamji AF, Clish CB, Brown LM, Girotti AW, Cornish VW, Schreiber SL, Stockwell BR. Regulation of ferroptotic cancer cell death by GPX4. *Cell.* 2014;156(1-2):317-331.

15. Sun QA, Kirnarsky L, Sherman S, Gladyshev VN. Selenoprotein oxidoreductase with specificity for thioredoxin and glutathione systems. *Proc Natl Acad Sci U S A.* 2001;98(7):3673-8.

16. Llabani E, Hicklin RW, Lee HY, Motika SE, Crawford LA, Weerapana E, Hergenrother PJ. Diverse compounds from pleuromutilin lead to a thioredoxin inhibitor and inducer of ferroptosis. *Nat Chem.* 2019;11(6):521-532.

17. Doll S, Freitas FP, Shah R, Aldrovandi M, da Silva MC, Ingold I, Goya Grocin A, Xavier da Silva TN, Panzilius E, Scheel CH, Mourão A, Buday K, Sato M, Wanninger J, Vignane T, Mohana V, Rehberg M, Flatley A, Schepers A, Kurz A, White D, Sauer M, Sattler M, Tate EW, Schmitz W, Schulze A, O'Donnell V, Proneth B, Popowicz

GM, Pratt DA, Angeli JPF, Conrad M. FSP1 is a glutathione-independent ferroptosis suppressor. *Nature*. 2019;575(7784):693-698.

18. Bersuker K, Hendricks JM, Li Z, Magtanong L, Ford B, Tang PH, Roberts MA, Tong B, Maimone TJ, Zoncu R, Bassik MC, Nomura DK, Dixon SJ, Olzmann JA. The CoQ oxidoreductase FSP1 acts parallel to GPX4 to inhibit ferroptosis. *Nature*. 2019;575(7784):688-692.

19. Soula M, Weber RA, Zilka O, Alwaseem H, La K, Yen F, Molina H, Garcia-Bermudez J, Pratt DA, Birsoy K. Metabolic determinants of cancer cell sensitivity to canonical ferroptosis inducers. *Nat Chem Biol*. 2020;16(12):1351-1360.

20. Shimada K, Skouta R, Kaplan A, Yang WS, Hayano M, Dixon SJ, Brown LM, Valenzuela CA, Wolpaw AJ, Stockwell BR. Global survey of cell death mechanisms reveals metabolic regulation of ferroptosis. *Nat Chem Biol*. 2016;12(7):497-503.

21. Shimada K, Hayano M, Pagano NC, Stockwell BR. Cell-Line Selectivity Improves the Predictive Power of Pharmacogenomic Analyses and Helps Identify NADPH as Biomarker for Ferroptosis Sensitivity. *Cell Chem Biol*. 2016;23(2):225-235.

22. Leu JI, Murphy ME, George DL. Functional interplay among thiol-based redox signaling, metabolism, and ferroptosis unveiled by a genetic variant of TP53. *Proc Natl Acad Sci U S A*. 2020;117(43):26804-26811.

23. Balsa E, Soustek MS, Thomas A, Cogliati S, García-Poyatos C, Martín-García E, Jedrychowski M, Gygi SP, Enriquez JA, Puigserver P. ER and Nutrient Stress Promote Assembly of Respiratory Chain Supercomplexes through the PERK-eIF2 α

Axis. *Mol Cell*. 2019;74(5):877-890.e6.

24. Ohashi K, Kawai S, Murata K. Identification and characterization of a human mitochondrial NAD kinase. *Nat Commun*. 2012;3:1248.

25. Ding CC, Rose J, Sun T, Wu J, Chen PH, Lin CC, Yang WH, Chen KY, Lee H, Xu E, Tian S, Akinwuntan J, Zhao J, Guan Z, Zhou P, Chi JT. MESH1 is a cytosolic NADPH phosphatase that regulates ferroptosis. *Nat Metab*. 2020;2(3):270-277.

Major issue 2: Furthermore, the authors should add a schematic diagram to summarize their proposed model, which will help reviewers and readers to better comprehend their suggested mechanism.

Response: We thank the reviewer's constructive suggestion. We did add a schematic diagram (Figure 8G) to summarize the proposed model, which helps reviewers and readers to better comprehend the suggested mechanism.

Major issue 3: They need to show western blotting data of SOX13 knockdown in cisplatin resistant cells (corresponding to Fig. S6E).

Response: We thank the reviewer's helpful suggestion. We have added the western blotting data of SOX13 knockdown in cisplatin resistant cells (corresponding to Fig. S6E). It was demonstrated in Fig. S6E in the revised version.

Reviewer #2:

The authors have adequately addressed my comments from the initial round of review.

Response: We thank the reviewer's recognition.

Reviewer #3:

Major issue 1: There are many important key points that are claimed but not substantiated. For example, (line 108) the authors write "GC is the second most insensitive tumor type to Erastin (Supplementary Figure 1A), suggesting that GC is relatively resistant to ferroptosis induction." How was this analysis performed, on what basis was this claim made, what are the numbers etc? Perhaps the data is presented in the Supplementary Information, but please do not expect a reader or a reviewer to flip back and forth between the different sections.

Response: We thank the reviewer's helpful suggestion. We have added "After analysis of the sensitivity of 20 tumor types to Erastin (a typical ferroptosis inducer) using the DepMap database, GC is the second most insensitive tumor type to Erastin (Supplementary Figure 1A), suggesting that GC is relatively resistant to ferroptosis induction" in the Results section.

Major issue 2: Likewise, there are many key facts that are not explained, making it very difficult for a reviewer to assess the reliability of the results. Specifically,

what is Erastin? What is RSL-3? The first time these compounds are mentioned on line 112 and it is unlikely a reader not intimately knowledgeable about ferroptosis will be familiar with these compounds (unlike say well-known genes like P53). Why were two different compounds used? Do the cell lines show cross-resistance?

Response: We thank the reviewer's helpful suggestion. We have added "Erastin is a strong inhibitor of system Xc⁻ and RSL3 is the prototypical GPX4 inhibitor. They work on different endogenous ferroptosis inhibitory systems; thus, we choose them for subsequent experiments" in the Introduction section.

Major issue 3: The association between high SOX13 expression and ferroptosis resistance should be validated in more cell lines that show endogenous sensitivity or resistance to ferroptosis.

Response: We thank the reviewer's constructive suggestion. We have added "The association between high SOX13 expression and ferroptosis resistance were validated in 786-O (human clear cell renal cell carcinoma) cell line, which shows heightened sensitivity to ferroptosis inducers. SOX13 over-expressing 786-O cells were less sensitive to ferroptosis inducers, including Erastin and RSL3 (Supplemental Figure 5D)."

Major issue 4: The specificity of the SOX13-associated resistance could be further

tested. It seems that high SOX13 imparts resistance to ferroptosis and cisplatin, but not apoptosis, doxorubicin, or gemcitabine. It would be useful to test a few more commonly used gastric cancer chemotherapies such as 5-FU, oxaliplatin, and taxanes.

Response: We thank the reviewer's constructive suggestion. We have added "A number of studies have demonstrated that ferroptosis is involved in the resistance of the commonly used chemotherapeutic agents 5-FU and oxaliplatin. We found that overexpression of SOX13 decreased the sensitivity to oxaliplatin and 5-FU in parental SNU-668 cells (Supplemental Figure 6F)." in the Results section.

Major issue 5: (line 203) "When we compared the commonly dysregulated genes detected by RNA-seq with SOX13 downregulation, the genes with SOX13-bound transcription sites characterized by ChIP-seq and the genes dysregulated in FIN-resistant cells, we identified one overlapping gene SCAF1 (Figure 3A)." This seems a bit strange to me in that the overlap of the three data sets only pinpoints one gene, and could this be random chance? This could be assessed in a permutation analysis.

Response: We thank the reviewer's constructive suggestion. Actually, it is actually the overlap of four data sets. When we compared the commonly dysregulated genes detected by RNA-seq with SOX13 downregulation, the genes with SOX13-bound transcription sites characterized by ChIP-seq and the genes dysregulated in FIN-resistant cells (Erastin resistant SNU-668 cells V.S. parental SNU-668 cells, RSL3

resistant SNU-484 cells V.S. parental SNU-484 cells). It was identified through a series of vigorous experimental validation. The correlation between SOX13 and SCAF1 was found both on expression and function.

Major issue 6: (line 225) “According to JASPAR, we identified three potential SOX13 binding motifs in the promoter area of SCAF1 (2000 bp upstream of the 226 transcription start site) (Figure 3G).” Do these three potential SOX13 binding motifs show SOX13 Chip-seq signals?

Response: We thank the reviewer’s constructive suggestion. Binding profiles and peak calling records of SOX13 in the SCAF1 promoter shows that the 300-bp region upstream of the transcription start site of SCAF1 was highly enriched for SOX13 binding (Supplemental Figure 10C).

Major issue 7: Is the extent of resistance conferred by SOX13 and SCAF1 exactly the same? It seems a bit surprising that among the many hundreds of genes SOX13 must regulate (as a transcription factor) that *all* of the effect goes through a single gene (SCAF1). It would be useful to compare the resistance/sensitivity of SCAF1 knockdown vs SANF1 + SOX13 knockdown.

Response: We thank the reviewer’s constructive suggestion. We found that the ferroptosis-resistance conferred by SOX13 could be attenuated by SCAF1 downregulation (Figure 4C, Supplemental Figure 14A-D). Actually, SCAF1 was

identified through overlapping of four data sets and validated through a series of vigorous experiments. The correlation between SOX13 and SCAF1 was found both on expression and function. SOX13 might have a variety of biological functions, however, its effect on ferroptosis could be largely achieved through the regulation of SCAF1. In order to compare the resistance/sensitivity of SCAF1 knockdown vs SCAF1 + SOX13 knockdown, we performed additional experiments. We found that the effect of SCAF1 knockdown was comparable to SCAF1 + SOX13 knockdown in terms of resistance.

Major issue 8: (line 316) “The increased NADPH production when acquiring ferroptosis-resistance was validated in vivo.” Does this refer to Figure 5I? I do not think this is strong rigorous evidence, as changes in blood could be due to a wide variety of causes.

Response: We thank and accept the reviewer’s constructive suggestion. We have

rephrased the sentence “The *in vivo* data also provided evidence to the increased NADPH production when acquiring ferroptosis-resistance.”

Major issue 9: Overall, I find that this study suffers from too many claims, each supported by a fairly thin line of evidence. The study would be better off being broken up into two manuscripts – perhaps as a suggestion one focused on the role of SOX13/SCAF1 in ferroptosis, and another on the therapeutic mechanisms (immune blockade, zanamivir and its mechanism of action).

Response: We thank the reviewer’s constructive suggestion. Our paper consists of two parts: one focused on the role and the relevant molecular mechanism of SOX13/SCAF1 in ferroptosis, the other part focused on the SOX13-targeting agent zanamivir and its role in ferroptosis. We appreciate the concern proposed by Reviewer 3 and are worried that the integrity and novelty might got jeopardized by broken up into two manuscripts.

Minor Comments

Minor issue 1: I would suggest the manuscript should undergo some copyediting – phrases like “Ferroptosis has been linked to the efficacy of a lot of anticancer therapies,” (line 96) would be better phrased as “multiple anticancer therapies”. This will improve readability.

Response: We thank and accept the reviewer’s constructive suggestion. We have revised accordingly.

Minor issue 2: (Line 275) “To further consolidate the clinical relevance of ferroptosis to cisplatin-based chemotherapy in GC patients, we performed 4-HNE staining in pre- and post-chemotherapy GC samples in cohort 2.” What is 4-HNE?

Response: We thank and accept the reviewer’s constructive suggestion. We have revised accordingly. “4-hydroxy-2-nonenal (4-HNE, a well-known by-product of lipid peroxidation and is widely accepted as a stable marker for oxidative stress)”.

Reviewers' Comments:

Reviewer #1:

Remarks to the Author:

The authors proposed a hypothesis regarding how SCAF1 suppresses ferroptosis, yet it appears that their model lacks coherence. While it is well-established that NADPH plays a crucial role in suppressing ferroptosis, the critical aspect lies in understanding how the electron transport chain (ETC) complex regulates NADPH levels. The authors suggested that NADH produced by the ETC complex is further converted to NADPH by NAD kinase. However, it is known that NADH is consumed, rather than produced, in the ETC complexes, as the electron from NADH is transferred to complex I, leading to the conversion of NADH to NAD⁺; thus, higher ETC function should result in decreased NADH levels.

According to their model, SCAF1 promotes the assembly of complex III and IV, thereby enhancing electron transport in the ETC. Consequently, SCAF1 deficiency would be expected to reduce mitochondrial OxPHOS activity (as demonstrated in the study) while increasing NADH levels (notably, NAD⁺ and NADH levels were not measured in this study). Fundamentally, it seems that the role of SCAF1 as an assembly factor for ETC complexes cannot account for their observation that NADPH levels are diminished in SCAF1-deficient cells.

Despite the authors' explanation of their hypothesis in the rebuttal letter—where SCAF1 regulates ETC complexes, ETC influences NADH levels, NADH is converted to NADPH by NADK, and finally, NADPH is utilized to suppress ferroptosis—it is evident that this mechanism lacks coherence, as pointed out by the reviewer. Furthermore, the authors made minimal changes to their corresponding text, leaving the logic unclear. Additionally, in the new Figure 8G, the schematic directly links ETC to NADPH, which is confusing, given that ETC does not directly affect NADPH levels.

"The effect of SOX13/SCAF1 on the production of NADH was confirmed in Supplemental Figure 16." This figure contains NADPH data, but not NADH data.

Overall, I agree with Reviewer 3 that even though this study presents a large amount of data, it lacks coherence in mechanistic studies and the readability is considered low.

Reviewer #3:

Remarks to the Author:

The amendments and additional data provided by the authors have definitely improved the flow of the manuscript. I have no further comments.